# Salivary proteins potentially derived from horizontal gene transfer are critical for salivary sheath formation and other feeding processes
Hai-Jian Huang [1,3] ✉, Li-Li Li[1,3], Zhuang-Xin Ye[1], Jia-Bao Lu[1], Yi-Han Lou[2], Zhong-Yan Wei[1], Zong-Tao Sun[1], Jian-Ping Chen [1], Jun-Min Li [1] & Chuan-Xi Zhang [1] ✉

Herbivorous insects employ an array of salivary proteins to aid feeding. However, the mechanisms behind the recruitment and evolution of these genes to mediate plant-insect interactions remain poorly understood. Here, we report a potential horizontal gene transfer (HGT) event from bacteria to an ancestral bug of Eutrichophora. The acquired genes subsequently underwent duplications and evolved through co-option. We annotated them as horizontal-transferred, Eutrichophora-specific salivary protein (HESPs) according to their origin and function. In *Riptortus pedestris* (Coreoidea), all nine HESPs are secreted into plants during feeding. The RpHESP4 to RpHESP8 are recently duplicated and found to be indispensable for salivary sheath formation. Silencing of RpHESP4-8 increases the difficulty of *R. pedestris* in probing the soybean, and the treated insects display a decreased survivability. Although silencing the other RpHESPs does not affect the salivary sheath formation, negative effects are also observed. In *Pyrrhocoris apterus* (Pyrrhocoroidea), five out of six PaHESPs are secretory salivary proteins, with PaHESP3 being critical for insect survival. The PaHESP5, while important for insects, no longer functions as a salivary protein. Our results provide insight into the potential origin of insect saliva and shed light on the evolution of salivary proteins.

Saliva is an oral secretion composed of a mixture of bioactive compounds that enable insects to feed and survive successfully. Analysis of salivary components from planthoppers, aphids, whiteflies, and true bugs has revealed abundant orphan genes that are restricted to a single species or a particular taxonomic group[1–3]. In the past decades, the functions of several orphan salivary proteins have been uncovered. For example, Te28 and Te84 from the spider mite were found to suppress salicylic acid (SA)-mediated plant defense[4]; Bsp9 from the whitefly was found to target host immunity regulators[5]; and LsSP1 from the planthopper was found to reduce elicitor-induced plant immunity[6]. Despite these findings, it remains unknown where these orphan salivary components originate and how they have evolved in the long-term coevolution of insects and plants.

The salivary sheath, formed by gel saliva, is critical for insect feeding. This unique structure is closely linked to the probing action of the stylet, serving multiple functions, from enhancing mechanical stability to lubricating stylets[7]. Disruption of salivary sheath formation negatively affects insect feeding from plant sieve tubes, while it does not impact insect feeding from artificial diets[8,9]. In planthoppers, salivary sheath-deficient insects fail to anchor their stylet at a fixed point, leading to a failure in penetration initiation and higher mortality[8]. The salivary sheath directly contacts host plants and plays important roles in herbivore–plant interactions. A few proteins in the salivary sheath were reported to exhibit a high evolutionary rate[10]. However, given the difficulty in collecting the salivary sheath and dissolving proteins, only a few salivary sheath proteins were identified, with the majority exhibiting species specificity[11].

[1]State Key Laboratory for Managing Biotic and Chemical Threats to the Quality and Safety of Agro-Products, Key Laboratory of Biotechnology in Plant Protection of Ministry of Agriculture and Zhejiang Province, Institute of Plant Virology, Ningbo University, Ningbo 315211, China. [2]Zhejiang Provincial Center for Disease Control and Prevention, Hangzhou 310051, China. [3]These authors contributed equally: Hai-Jian Huang, Li-Li Li. ✉e-mail: huanghaijian@nbu.edu.cn; chxzhang@zju.edu.cn

Horizontal gene transfer (HGT), the transmission of genetic material between distantly related species, is considered as the major driving force in life evolution[12]. In recent years, advancements in high-throughput sequencing have shown the prevalence and importance of HGTs, particularly in bacteria and protists, which have close associations with their metazoan recipients[13,14]. Some of these transferred genes have been found to confer direct benefits to their recipients, such as LOC105383139 gene for male courtship[15], carotenoid biosynthesis gene for aphid body coloration[16], and parasitoid killing factor for insect defense to parasitoids[17]. HGT also plays critical roles in shaping the plant-insect interaction. Accumulated evidence demonstrated that HGT from microbial species enhanced the enzymatic repertoire of herbivorous insects, which in turn facilitated their adaptation toward herbivory and novel host plants[18]. For example, the widespread of horizontally transferred polygalacturonases, cellulases, and xylanases in insect genome provided selective advantage to recipients by degrading plant cell wall components[19,20], while the β-fructofuranosidase, cysteine synthase, and biotin synthase involved in assimilation of plant nutrients[21–23]. In addition, insects have evolved various counter adaptations to plant defenses by acquiring pre-evolved genes from bacteria and fungi[18]. In *Bemisia tabaci*, a fungal transferred salivary protein was reported to be migrated into plant cells and promoted insect feeding by suppressing ferredoxin-triggered plant immunity[24]. Generally, the majority of reported horizontally transferred genes that were associated with insect feeding code for enzymes, and function similarly between the donor and recipient organisms[18]. Our knowledge on their evolved function after co-opted by the insect recipient is still limited.

Pentatomomorpha, an infraorder of Heteroptera, encompasses five superfamilies. Among these, three superfamily of Coreoidea, Lygaeoidea, and Pyrrhocoroidea, grouped as Eutrichophora, comprise more than 8500 described species, and are paraphyletic to a significant extent[25]. *Riptortus pedestris* (Coreoidea), *Pyrrhocoris apterus* (Pyrrhocoroidea), and *Oncopeltus fasciatus* (Lygaeoidea) are important species in these three superfamilies with available genome information[26,27]. Our previous study revealed that *R. pedestris* contains abundant species-specific genes with unknown origins and functions[26]. In this study, we provide evidence for the potential horizontal transfer of a bacterial gene to the ancestral of Coreoidea, Lygaeoidea, and Pyrrhocoroidea. The acquired genes undergo duplications, gain introns, and evolve through co-option in recipient insects and serve as the main salivary components that enhance insect feeding.

## Results

### Identification of potential horizontally transferred salivary proteins in *R. pedestris*

The salivary proteins of *R. pedestris* were identified by comparing the untreated soybeans and soybeans infested with *R. pedestris* using shotgun LC − MS/MS analysis. Out of the 40 salivary proteins that were identified (Table 1), 9 displayed Eutrichophora-specificity and showed sequence similarity to bacterial proteins. We therefore annotated them as horizontal-transferred, Eutrichophora-specific salivary protein (HESPs) (Fig. 1a, Supplementary Table 1). Noteworthy, these nine salivary RpHESPs displayed 15.4% to 98.0% amino acid sequence similarity to each other, with a hit e-value ranging from 0 to $10^{-24}$ (Fig. 1b, Supplementary Fig. 1). To identify additional RpHESP homologs in *R. pedestris*, the nine RpHESPs were searched against the *R. pedestris* genome. In addition to these nine RpHESPs, we did not find other RpHESPs. These results suggest that all RpHESPs in *R. pedestris* are secreted into plants during insect feeding.

The nine RpHESPs were tandemly arranged on Chromosome 1, and we designate them as RpHESP1 to RpHESP9 based on their genomic position (Fig. 2a). The RpHESP2 to RpHESP9 genes were located in adjacent gene loci and shared the same transcription orientations, while RpHESP1 was independently located and had different transcription orientations from the other RpHESPs. Except for RpHESP2 that contained 13 exons, the other RpHESPs contained 3 exons (Fig. 2a). To validate the insect origin of these RpHESP sequences, we further analyzed the flanking genes of RpHESPs in the genome. The results showed that the flanking

genes upstream and downstream of RpHESPs were belonged to insects (Supplementary Table 2). In addition, the flanking region between RpHESP1 and an insect gene was successfully amplified by PCR, indicating the accuracy of genome assembly. These results suggest that RpHESPs are integrated in the insect genome, but not derive from bacteria or other contamination.

All RpHESP genes have a signal peptide sequence, indicating their potential to be secreted (Fig. 1a). Then, the spatial-temporal expression patterns of RpHESPs in different tissues and developmental stages were analyzed. According to our results, all the RpHESP genes showed similar expression patterns in tissues and development stages, respectively (Fig. 1c, Supplementary Fig. 2). With regard to developmental analysis, the newly deposited eggs expressed the lowest levels of RpHESP transcripts. Since then, the transcript levels gradually increased along with egg development, and remain high in the nymph and adult stages (Supplementary Fig. 2). In tissue analysis, all RpHESP transcripts were found to be expressed exclusively in the salivary glands (Fig. 1c).

### HESPs are potentially transferred from bacteria to an ancestral bug

BLASTing RpHESPs against the NCBI nr database revealed that the RpHESPs had no sequence similarity to any known proteins in eukaryotes, with the exception of an uncharacterized protein found in the milkweed bug *O. fasciatus* (Lygaeoidea). To gain a comprehensive understanding of HESP phylogeny, transcriptomic data from 60 insect species (representing 14 superfamilies within Hemiptera) were analyzed. With the exception in *Anoplocnemis dalasi* (Coreoidea), the HESP-associated contigs were found in almost all species in Eutrichophora (Supplementary Table 3). In contrast, no HESP homologs were detected in other insect species. An evolutionary analysis based on 34 single copy genes demonstrated that Eutrichophora species (Coreoidea, Lygaeoidea, and Pyrrhocoroidea superfamilies) clustered in the same clade and that Eutrichophora diverged from the Pentatomoidea superfamily at ~117–141 million years ago (Mya) (Fig. 3). Given that HESP distributed in nearly all Eutrichophora species, but not the Pentatomoidea species (Supplementary Table 3), we speculated that HESP might be evolved ~117–141 Mya, the time during the divergency of Eutrichophora and Pentatomoidea.

The genomes of *R. pedestris*, *P. apterus*, and *O. fasciatus* are currently available for analysis. In *P. apterus*, six PaHESPs were identified and were found to be tandemly arranged in adjacent gene loci on Scaffold 25, suggesting the possibility of duplication events (Fig. 2b). PaHESPs consisted of 2-4 exons. PaHESP5 exhibited different transcriptional orientation compared to the other PaHESPs (Fig. 2b). In *O. fasciatus*, four OfHESPs were identified. Two OfHESPs were arrayed in adjacent gene loci on Scaffold 5, while the other two were located on Scaffold 45 and Scaffold 607, respectively (Fig. 2c).

All HESPs in the three genome-available insects displayed sequence similarity to bacterial proteins. For example, RpHESP1, PaHESP1, and OfHESP1 were most closely related to genes from the *Omnitrophica bacterium*, with a hit e-value < $10^{-110}$. It is worth to mention that there are only 14 bacteria species contain HESP-associated genes, and except for these bacteria, no gene in fungi or other protists showed sequence similarity to insect HESPs. A maximum likelihood tree was then constructed based on the HESPs in the three insects and HESP-associated genes in bacteria (Fig. 4). The results showed that all insect HESPs were clustered in the same group, while the bacteria genes clustered together. RpHESP1, PaHESP1, and OfHESP1 showed high sequence similarity and were clustered in the same clade. RpHESP2 and PaHESP2 were clustered together in another clade, but no HESP gene in *O. fasciatus* was found in this clade. It is possible that RpHESP2 and PaHESP2 evolved after the divergence of Lygaeoidea from Pyrrhocoroidea and Coreoidea (Fig. 3), or because of the independent gene losses in *O. fasciatus*. Our study also identified relatively recent gene duplication events in each of the three species; for example, RpHESP4 to RpHESP8 had more than 90.2% amino acid similarity (Fig. 4; Supplementary Fig. 1). These results

**Table 1 | Identification of salivary proteins in *Riptortus pedestris*[a]**

| Protein ID | Description | Signal peptide | Salivary gland-specific | Peptide counts | Unique peptides | Coverage [%] | MW [kDa] | LC-MS/MS None-infested | Infested R1 | Infested R2 |
|---|---|---|---|---|---|---|---|---|---|---|
| Chr1.0517 | RpHESP1 | SP | Y | 63 | 63 | 62.6 | 113 | - | + | + |
| Chr2.0916 | Uncharacterized protein | - | Y | 51 | 51 | 19.4 | 483 | - | + | + |
| Chr2.2953 | Uncharacterized protein | SP | Y | 25 | 25 | 26.4 | 146.3 | - | + | + |
| Chr1.0276 | RpHESP4 | SP | Y | 21 | 6 | 54.4 | 72.5 | - | + | + |
| Chr1.0272 | RpHESP7 | SP | Y | 20 | 7 | 57.4 | 66.6 | - | + | + |
| Chr1.0271 | RpHESP8 | SP | Y | 20 | 2 | 55.6 | 67 | - | + | + |
| Chr1.0274 | RpHESP6 | SP | Y | 20 | 4 | 60.4 | 66.4 | - | + | + |
| Chr1.0275 | RpHESP5 | SP | Y | 19 | 2 | 57.4 | 67.1 | - | + | + |
| Chr1.0278 | RpHESP2 | SP | Y | 14 | 14 | 34.4 | 55.5 | - | + | + |
| Chr1.0277 | RpHESP3 | SP | Y | 11 | 11 | 23.4 | 80.7 | - | + | + |
| Chr1.0270 | RpHESP9 | SP | Y | 11 | 11 | 19.6 | 72.5 | - | + | + |
| Chr1.4000 | Uncharacterized protein | SP | Y | 9 | 9 | 36.8 | 46.6 | - | + | + |
| Chr3.2367 | Venom serine protease | SP | Y | 9 | 9 | 31.5 | 35.5 | - | + | + |
| Chr1.0508 | Uncharacterized protein | SP | Y | 8 | 8 | 44.4 | 26.4 | - | + | + |
| Chr1.3998 | Uncharacterized protein | SP | Y | 8 | 8 | 14.7 | 59 | - | + | + |
| Chr5.1560 | Inter-alpha-trypsin inhibitor | SP | - | 8 | 8 | 12.4 | 92.9 | - | + | + |
| Chr1.1122 | Conserved secreted protein | SP | Y | 7 | 7 | 65.9 | 13.8 | - | + | + |
| Chr4.2119 | Uncharacterized protein | - | Y | 7 | 7 | 41.9 | 28 | - | + | + |
| Chr1.2764 | Uncharacterized protein | SP | Y | 5 | 5 | 44.9 | 15.6 | - | + | + |
| Chr4.0863 | Uncharacterized protein | - | Y | 5 | 5 | 18.2 | 26.7 | - | + | + |
| Chr5.1278 | Uncharacterized protein | - | - | 5 | 5 | 45.3 | 14.6 | - | + | + |
| Chr2.1727 | Uncharacterized protein | SP | - | 4 | 4 | 24.8 | 25 | - | + | - |
| Chr4.1618 | Uncharacterized protein | SP | Y | 4 | 4 | 23.6 | 26 | - | + | + |
| Chr1.3989 | Uncharacterized protein | SP | Y | 4 | 4 | 14.6 | 37.4 | - | + | + |
| Chr4.1608 | Uncharacterized protein | SP | Y | 4 | 4 | 22.1 | 30.5 | - | + | + |
| Chr4.1616 | Uncharacterized protein | SP | Y | 4 | 4 | 22 | 28 | - | + | + |
| Chr1.4144 | Uncharacterized protein | SP | Y | 3 | 3 | 9.8 | 397.1 | - | + | + |
| Chr4.1617 | Uncharacterized protein | SP | Y | 3 | 3 | 17.5 | 22.7 | - | + | + |
| Chr4.1609 | Uncharacterized protein | SP | Y | 3 | 3 | 11.9 | 48.7 | - | - | + |
| Chr1.0507 | Uncharacterized protein | SP | Y | 2 | 2 | 24.6 | 18.2 | - | + | + |
| Chr1.0509 | Uncharacterized protein | SP | Y | 2 | 2 | 13.2 | 39 | - | + | - |
| Chr1.1506 | Chymotrypsin-1 | - | Y | 2 | 2 | 6.2 | 55.5 | - | + | + |
| Chr1.3987 | Uncharacterized Protein | SP | Y | 2 | 2 | 12.2 | 25 | - | + | + |
| Chr1.3991 | Uncharacterized protein | SP | Y | 2 | 2 | 9.8 | 31.8 | - | + | - |
| Chr4.4166 | Uncharacterized protein | SP | Y | 2 | 2 | 17.8 | 16 | - | + | + |
| Chr2.0785 | Uncharacterized protein | SP | Y | 2 | 2 | 36.5 | 10.3 | - | + | + |

https://doi.org/10.1038/s42003-024-05961-9 **Article**

**Table 1 (continued) | Identification of salivary proteins in *Riptortus pedestris*[a]**

| Protein ID | Description | Signal peptide | Salivary gland-specific | Peptide counts | Unique peptides | Coverage [%] | MW [kDa] | LC-MS/MS | | |
|---|---|---|---|---|---|---|---|---|---|---|
| | | | | | | | | None-infested | Infested R1 | Infested R2 |
| Chr2.1734 | Uncharacterized protein | SP | - | 2 | 2 | 5.9 | 47.6 | - | + | - |
| Chr4.0644 | Zinc finger protein | - | - | 2 | 2 | 2.4 | 123.5 | - | + | - |
| Chr4.2571 | Uncharacterized protein | SP | Y | 2 | 2 | 13.3 | 24.8 | - | + | + |
| Chr1.0072 | Proteasome | - | - | 2 | 2 | 7.6 | 31.8 | - | + | + |

[a]The expression patterns of each gene in salivary glands, guts, carcass, testes, and ovaries were determined by transcriptomic sequencing. "Y" indicated genes with fold changes>10 in all comparison between salivary glands and other tissues. Details in Transcripts per million (TPM) expression values of each gene was available in Supplementary Data 1. "SP" indicated the presence of signal peptide. "+" indicated the presence of protein in the sample, while "—" indicated the absence.

suggest that HESPs are duplicated and separated into different clades after acquirement.

### Diversified roles of RpHESPs in *R. pedestris*

RNAi experiments were performed to evaluate the potential functions of RpHESPs. As RpHESP4 to RpHESP8 shared >92% nuclear acid sequence similarity (Supplementary Fig. 3), and it is difficult to silence these genes specifically. Therefore, RpHESP4 to RpHESP8 were collectively silenced using the conserved region (refer to RpHESP4-8). First, dsRNA targeting *RpHESP1*, *RpHESP2*, *RpHESP3*, *RpHESP4-8*, and *RpHESP9* were individually injected into third instar nymphs. Four days after the injection, the RNAi efficacy was evaluated, and the results showed that the dsRNA specifically and effectively reduced the expression of the target genes, respectively (Supplementary Fig. 4). It is noteworthy that the expression of *RpHESP4-8* and *RpHESP9* were significantly induced when knocked down *RpHESP1* or *RpHESP2*, indicating the potential compensatory effect. After dsRNA inhibition, decreased survival was observed in *R. pedestris* treated with ds*RpHESP1*, ds*RpHESP3*, ds*RpHESP4-8*, and ds*RpHESP9*, while no significant difference in survival was observed between *R. pedestris* treated with ds*GFP* and ds*RpHESP2* (Fig. 5a). The most pronounced lethal effects were observed in insects treated with ds*RpHESP9*, with 56.4% died within 15 days, followed by ds*RpHESP3* (39.5%) and ds*RpHESP4-8* (29.7%).

During the feeding process, hemipteran insects usually secrete two types of saliva, gel saliva and watery saliva, into plant tissues. The gel saliva solidifies quickly and forms a salivary sheath that provides mechanical stability, lubrication, and protection against chemical defense[7]. Salivary sheath is an important indicator for insect feeding activity, as it always envelops stylets and is left in plant tissues after stylet withdrew[28]. We first dissected salivary sheath left in soybean seeds, and measured their length. The results showed that ds*RpHESP4-8*-treated *R. pedestris* secreted significant shorter salivary sheath than that of the ds*GFP*-treated control (Fig. 5b). In contrast, no significantly difference in length was observed in other ds*RpHESP*s treatments (Fig. 5b). The salivary sheath secreted by dsRNA-treated *R. pedestris* was also inspected by scanning electron microscopy. The salivary sheath secreted by ds*GFP*-treated controls were tubule-shaped with a hollow lumen that allowed stylet movement (Fig. 5c). Similar morphology was observed in salivary sheath secreted from ds*RpHESP1*-, ds*RpHESP2*-, ds*RpHESP3*-, and ds*RpHESP9*-treated *R. pedestris*. However, the tubular structure was collapsed in salivary sheath secreted by ds*RpHESP4-8*-treated *R. pedestris* (Fig. 5c), indicating the potential involvement of RpHESP4-8 in salivary sheath formation. To further investigate the relationship of RpHESPs and salivary sheath, the salivary sheath component was analyzed using LC-MS/MS analysis. The tubular structure outside the soybean seeds were cut and washed to remove potential watery saliva. As a result, all RpHESPs can be abundantly identified in salivary sheath (Supplementary Table 4). These results suggest that all RpHESPs are present in salivary sheath, but only RpHESP4-8 involves in maintaining the structure of the salivary sheath.

Thereafter, the insect probing frequency and probing time were monitored. The probing frequency was recorded by counting the salivary sheath left in soybean seeds, as *R. pedestris* secrete salivary sheath in each probing attempt. The results showed that the ds*RpHESP4-8*- and ds*RpHESP9*-treated *R. pedestris* probed less frequently than that of the ds*GFP* control (Fig. 5d). The probing time was recorded by measuring the time from probing initiation to successful penetration. The ds*GFP*-treated *R. pedestris* quickly arched their stylets in a fixed point and started penetration as fast as 3 min (Fig. 5e). Similar probing behaviors were observed in ds*RpHESP1*-, ds*RpHESP2*-, and ds*RpHESP3*-treated *R. pedestris*. In contrast, the ds*RpHESP4-8*- and ds*RpHESP9*-treated insects repeatedly attempted to probe the soybean but failed to start penetration (Fig. 5e). The mean time for ds*RpHESP4-8*- and ds*RpHESP9*-treated *R. pedestris* to initiate penetration was 12 min and 10 min, respectively, which were significantly longer than that of ds*GFP*-treated insects (Fig. 5e). These results suggested that ds*RpHESP4-8*- and ds*RpHESP9*-treatment have negative effect on insect probing process.

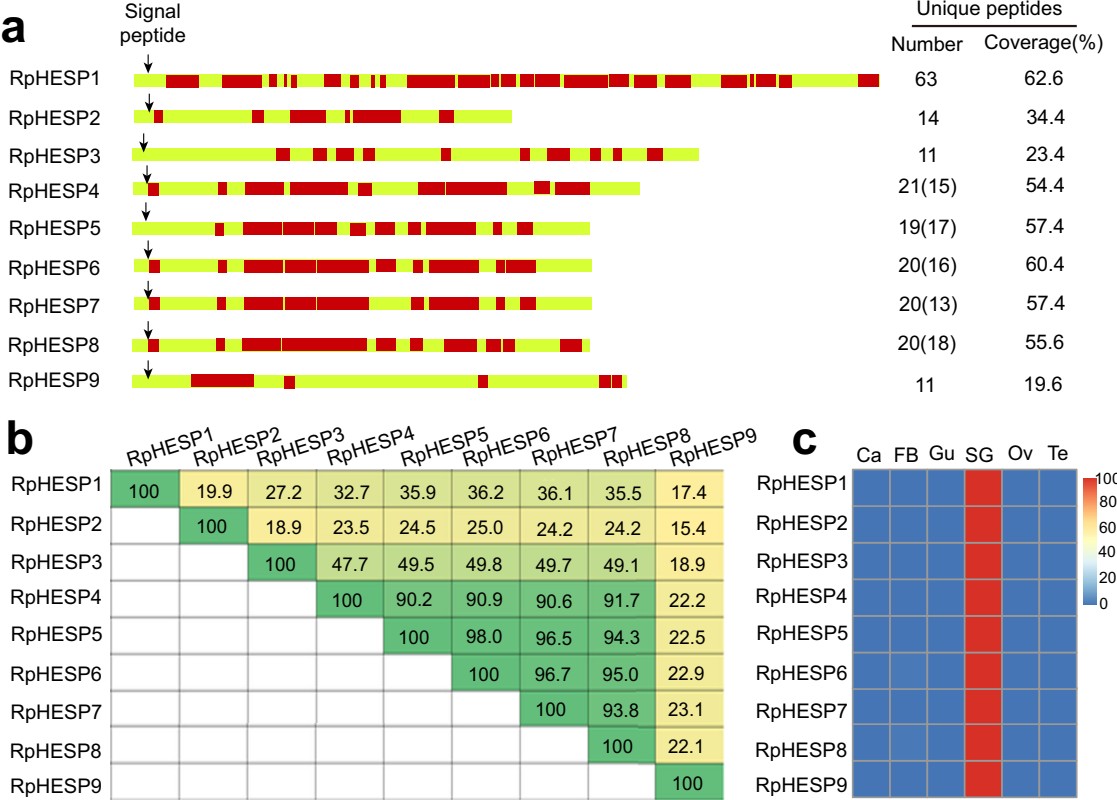

**Fig. 1 | Identification of RpHESPs in *Riptortus pedestris* saliva. a** Distribution of unique peptides identified by LC-MS/MS in *R. pedestris*-infested soybeans. Peptides that were mapped to RpHESPs are labeled in red. The number and coverage of unique peptides in each protein are displayed to the right. The number in brackets indicates unique peptides that belong to more than one protein. The arrow indicates the signal peptide cleavage site. **b** Comparison of amino acid identity among RpHESPs. The pairwise distances between the amino acid sequences of nine RpHESPs were calculated using MegAlign. **c** Expression patterns of RpHESPs in six tissues. Ca, carcass; FB, fat body, Gu, guts; SG, salivary glands; Ov, ovaries; Te, testis. The relative expression levels of RpHESPs were calculated based on qRT-PCR data and illustrated by a heat map.

## Diversified roles of PaHESPs in *P. apterus*

In this study, the function of HESP was further analyzed in the firebug *P. apterus*, a true bug mainly feeding on mallow, linden, and clover seeds. Although adjacently arrayed in the genome, these six PaHESP showed great variation in nucleic acid sequence length (from 972 bp to 4296 bp) and amino acid sequence similarity (from 19.8 to 83.8%) (Fig. 6a; Supplementary Fig. 5). The expression patterns of PaHESPs in different tissues were quantified by qRT-PCR. As a result, five out of six PaHESPs were specifically expressed in salivary glands, except for PaHESP5, which was prevalently expressed in salivary glands, carcass, testis, and ovaries (Fig. 6b). All of the PaHESPs were predicted to have a signal peptide, indicating a secretory nature (Fig. 6c). LC-MS/MS was used to determine if PaHESPs were secreted into clover seeds during feeding. Untreated clover seeds were used as a control. Five PaHESPs that specifically expressed in salivary glands were found to be secreted into the host in high amounts, with sequence coverage ranging from 47.4% to 62.3% and unique peptides ranging from 14 to 62 (Fig. 6c; Supplementary Table 5). By contrast, no PaHESP5-derived peptide was detected, suggesting that PaHESP5 no longer serves as a salivary protein after evolution.

RNAi experiments were performed to study the functions of PaHESPs. The dsRNA targeting *PaHESP1*, *PaHESP2*, *PaHESP3*, *PaHESP4*, *PaHESP5*, and *PaHESP6* were individually injected into third instar nymphs, respectively. qRT-PCR results demonstrated that the dsRNA can specifically and effectively reduce the expression of the target genes, respectively (Supplementary Fig. 6). Following dsRNA inhibition, decreased survivability was observed in ds*PaHESP1*-, ds*PaHESP3*- and ds*PaHESP5*-treated *P. apterus*. On the contrary, no significant difference in survival was found between ds*GFP* and other ds*PaHESP*-treated *P. apterus* (Fig. 6d). The salivary sheath secreted by ds*PaHESP*-treated *P. apterus* was then examined under

scanning electron microscopy. No noticeable change in the morphological structure of the salivary sheath was observed after ds*PaHESP* (Supplementary Fig. 7), indicating that no HESP is critical for maintaining the structure of the salivary sheath in *P. apterus*.

## Discussion

Salivary secretion in herbivorous insects is abundant in orphan proteins of unknown origin and function. This study documented a potential HGT event that occurred at ~117–141 Mya. After being transferred to insects, HESPs underwent duplications, gained introns, and evolved through co-option, allowing them to perform various functions in insect feeding and other processes. Our findings shed new light on the potential origin of orphan proteins in insects and suggest the potential evolution of salivary proteins.

The horizontal transferred genes in insects usually display two unusual properties: an absence of homologies in the majority of other lineages within the Insect phylum, and a high degree of sequence similarity to distantly related species, especially those from the protists[12,18]. In this study, HESPs was confirmed to be integrated into insect genome (Fig. 2; Supplementary Table 2), strictly distributed in three insect superfamily (Fig. 3; Supplementary Table 3), and showed high sequence similarity to bacteria genes (Fig. 4), suggesting a potential HGT origin. However, compared with most reported HGT events that had a lot of homologues in donor relatives, only a few bacteria species contain HESP-associated genes. Moreover, in current phylogenic tree, insect HESPs did not cluster with bacterial genes in strongly supported bootstrap (Fig. 4). Therefore, we cannot rule out the possibility that HESP-associated genes in bacteria and insects were independently evolved, and their sequence similarity might be resulted from convergent evolution. Also, there is a possibility that the HGT event was originated from

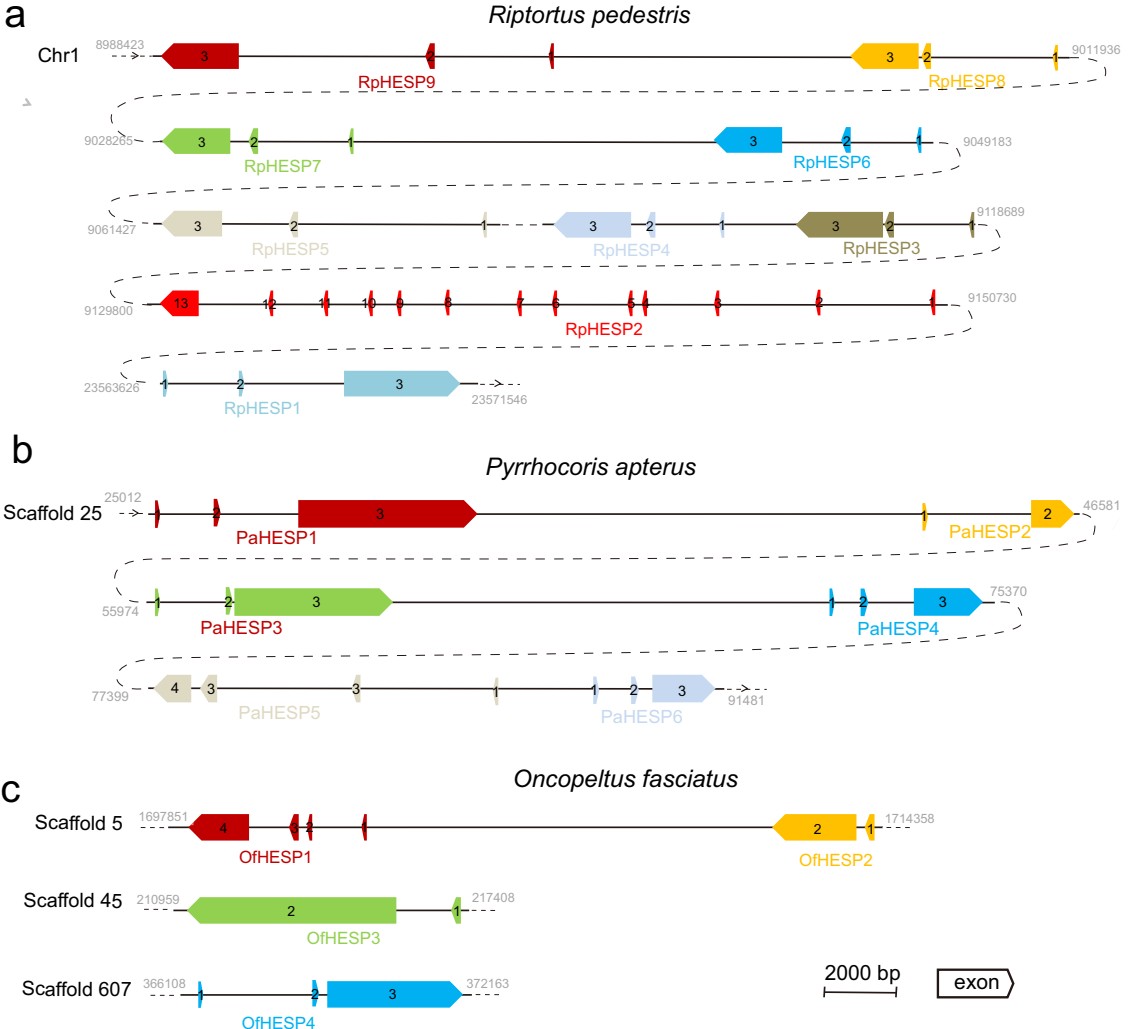

**Fig. 2 | Structures and locations of HESPs in three insect genomes.** The structures and locations of RpHESPs (**a**), PaHESPs (**b**), and OfEHESPs (**c**) in *Riptortus pedestris*, *Pyrrhocoris apterus*, and *Oncopeltus fasciatus* genomes were displayed. The boxed arrows indicate gene transcription orientations. By using the online tool Spidey, the coding sequences of HESPs were matched to genomic sequences to identify the exons and introns. The flanking genes of these HESPs in the genome were analyzed and displayed in Supplementary Table 3, which validated that the HESP-contained scaffolds/chromosomes were belong to insects, but not the bacterial contaminants.

insects to bacteria, as Eutrichophora species were intimately associated with symbiotic microorganism[29]. While our current evidence suggests HGT as the most likely scenario, additional work is needed to definitively rule out alternative hypotheses. Perhaps a high-throughput sequencing of a wider range of organisms in the future might solve this issue.

Recent advancements in HGT detection have shown that many apparent gene duplications are actually the result of HGT instead of autochthonous gene duplication[12,30]. Our study identified 9, 6, and 4 HESP genes in *R. pedestris*, *P. apterus*, and *O. fasciatus* genomes, respectively. The PaHESPs were found to be located adjacent to each other in the genome of *P. apterus* (Fig. 2b), suggesting the possibility of a single HGT event, followed by gene duplications. In contrast, although all RpHESPs in *R. pedestris* located in the same chromosome, RpHESP1 was ~$1.5 \times 10^7$-bp far away from the other RpHESPs and had different transcription orientations (Fig. 2a). Phylogeny analysis demonstrated that RpHESP1 was clustered with PaHESP1 and OfHESP1, but not the other RpHESPs (Fig. 4). Therefore, we cannot rule out the possibility that RpHESP1 and the other RpHESPs derived from two independent HGT events, which deserved further investigations.

Although most HESPs in *R. pedestris* and *P. apterus* likely evolved from autochthonous gene duplication, their functions varied greatly. For example, the newly duplicated RpHESP4 to RpHESP8 were crucial for maintaining the structure of the salivary sheath, while the other RpHESPs were not (Fig. 5). To sustainably survive in the recipient organism, a transferred gene typically needs to provide a selective advantage, either to itself or the recipient[31]. RNAi experiments demonstrated that multiple HESPs were important for insect survival (Figs. 5; 6), indicating that these genes underwent co-option evolution after being transferred to insect recipients. Despite being abundant in the secretion of RpHESP2, PaHESP2, PaHESP4, and PaHESP6 into plants, no significant phenotypic changes were observed in insects treated with corresponding dsRNA (Figs. 5; 6). *R. pedestris* and *P. apterus* are polyphagous insects that feed on a wide range of hosts or the same host at different developmental stages[26,32]. It is highly possible that some salivary proteins play a role in insects feeding on specific diets, which deserves further investigation. In addition, some horizontally transferred genes displayed functional redundancy after gene duplication in recipients[33,34]. There may have functional compensation in duplicated HESPs, which contribute to the genetic robustness and provided a selective advantage[35].

Although the RpHESP2 to RpHESP9 and PaHESP1 to PaHESP6 were derived from gene duplication, a low level of amino acid sequence similarity

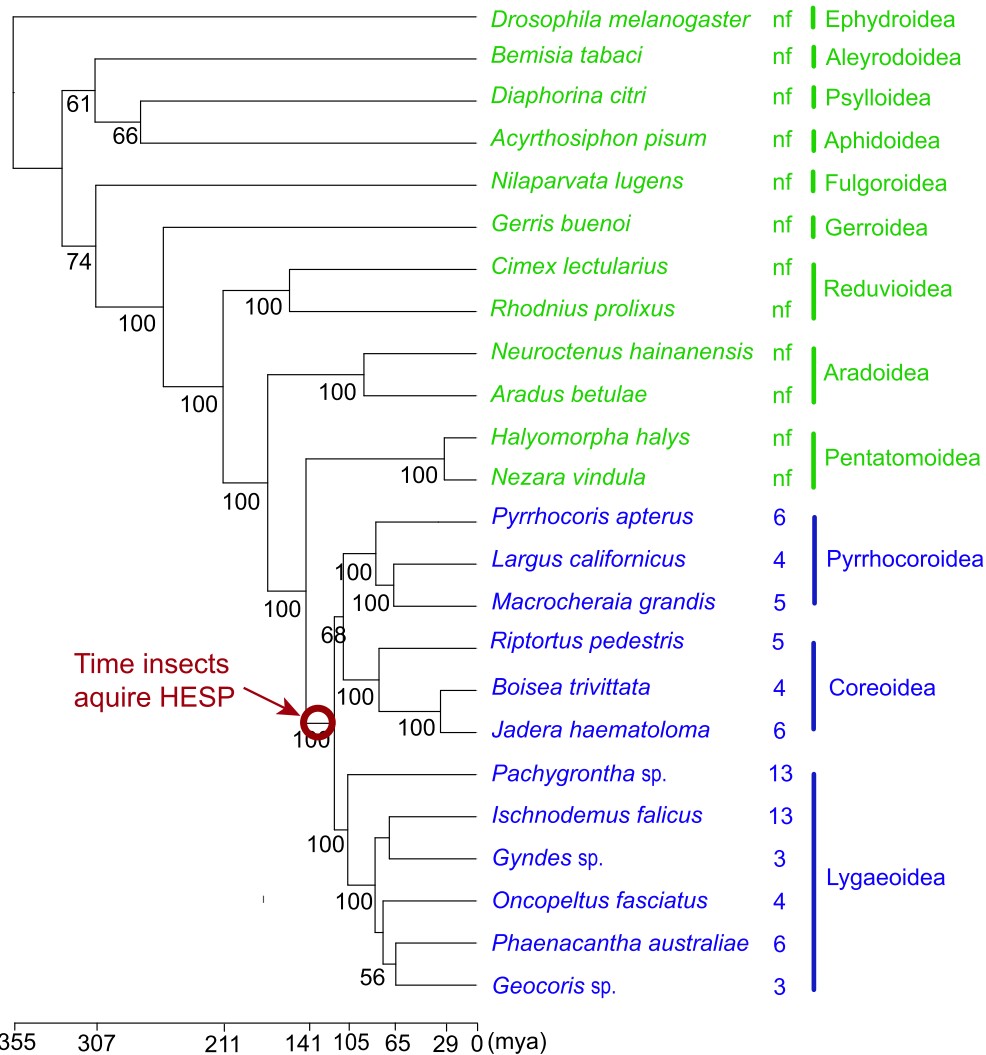

**Fig. 3 | Distribution of HESPs in hemipteran insects.** Representative insects from 12 different superfamilies were selected and a phylogenetic tree was constructed using 34 single-copy genes. *Drosophila melanogaster* was used as the root of the tree and the estimated species divergence time is shown at the bottom. The number of HESP-related contigs for each species is indicated on the right, with species with and without HESP-related contigs marked in blue and green, respectively. The best-fit amino acid substitution model was determined using ModelTest-NG52 and maximum likelihood trees were constructed using RAxML-NG with 1000 bootstrap replicates. The bootstrap proportions are indicated near the branches.

was found in some comparison groups, such as a 15.4% similarity between RpHESP2 and RpHESP9, and a 19.8% similarity between PaHESP1 and PaHESP3 (Supplementary Figs. 1, 5). Insect saliva plays critical roles in herbivore–plant interactions, and a few salivary proteins have been found to evolve at a high rate[10]. While it is likely that there are other HGT-derived orphan salivary proteins in insects, their donor genes have not yet been identified. This could be due to a lack of information about potential donors or the fast evolution of transferred genes in recipients, causing them to show no sequence similarity to their donors[12].

Overall, our study provides a comprehensive insight into the evolution of orphan salivary proteins (Supplementary Fig. 8). The HESPs were potentially transferred from bacteria to an ancestral bug of Coreoidea, Lygaeoidea, and Pyrrhocoroidea superfamily. These acquired genes underwent duplications, gained introns, and evolved through co-option, providing a selective advantage to the recipients by enhancing their feeding abilities or other undefined processes.

## Methods
### Insects
The *R. pedestris* strain originally collected from a soybean field (33.7° N, 117.0 °E) in Suzhou, China, was reared on soybean pods and soybean seeds

for more than 20 generations. The *P. apterus* strain was originally collected from Xinjiang, China, and reared on clover seeds. Both strains were maintained at 26 ± 0.5 °C and 50 ± 5% relative humidity under a light/dark photoperiod of 16/8 h.

### Collection of entire salivary secretion and salivary sheath
Before the experiment, both *R. pedestris* and *P. apterus* were subjected to a 24-h starvation period with only water being provided. Next, the insects were introduced into separate cages, one containing soybean seeds and the other clover seeds, and allowed to feed for 24 h. The presence of a salivary sheath indicated the feeding site of the insects, which can be easily observed under a SZ2-ILST Stereomicroscope (Olympus, Shinjuku, Japan). To analyze the component of entire salivary secretion, the salivary sheath and the surrounding area near the salivary sheath were carefully collected using a pair of forceps (Ideal-Tek, Balerna, Switzerland). To analyze the component of salivary sheath only, the tubular structure outside the soybean seeds were cut, and rinsed in 1× PBS to remove potential watery saliva remained in the salivary sheath. The insect salivary secretions were characterized by LC-MS/MS analysis. Two independent biological replicates were performed for each treatment. The none-infested seeds were used as a negative control.

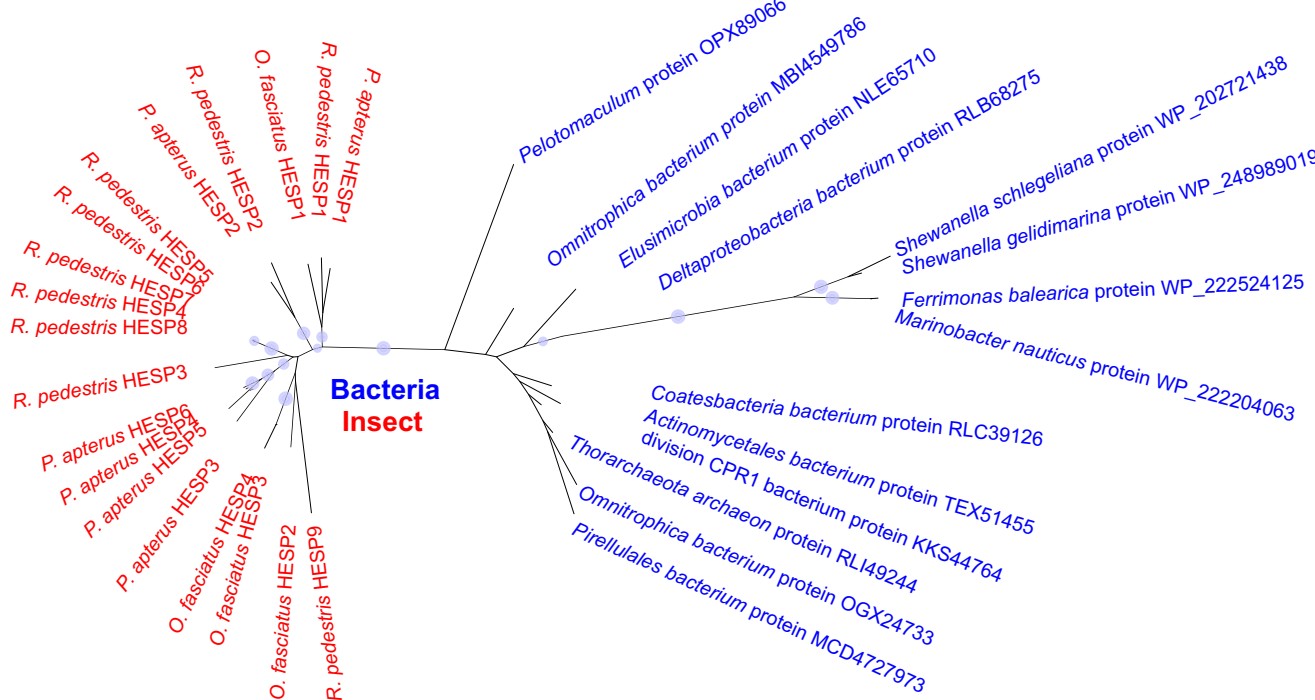

**Fig. 4 | Phylogeny analysis of insect HESPs and bacteria proteins.** The unrooted phylogenetic tree of HESP proteins in *Riptortus pedestris*, *Pyrrhocoris apterus*, and *Oncopeltus fasciatus* (red), along with close-related proteins from bacteria (blue), was constructed. The best-fit amino acid substitution model was determined using ModelTest-NG52 and maximum likelihood trees were constructed using RAxML-NG with 1000 bootstrap replicates. Nodes with bootstrap values > 60% are marked with solid blue circles, and the larger circles indicate higher bootstrap values.

## LC-MS/MS analysis

The collected samples were homogenized in the lysis solution containing 1% SDS, 100 mM DTT, and 100 mM Tris-HCl (pH 6.8), and then boiled for 10 min. Later, shotgun-based proteomic analysis was performed via the custom service of Novogene Institute (Novogene, Beijing, China) as previously described[2]. Briefly, the samples were lysed with a lysis buffer containing 100 mM NH$_4$HCO$_3$, 8 M Urea, and 0.2% SDS, followed by 5 min of ultrasonication on ice. Afterward, the lysate was centrifuged at 12,000 × $g$ for 15 min at 4 °C, and the resulting supernatant was transferred to a clean tube. The sample was then subjected to reduction using 10 mM DTT for 1 h at 56 °C, followed by alkylation with iodoacetamide in the dark at room temperature for 1 h. Subsequently, the sample was thoroughly mixed with 4 times its volume of pre-cooled acetone through vortexing and then incubated at −20 °C for at least 2 h. Following centrifugation, the precipitate was collected. After two washes with cold acetone, the pellet was dissolved in the dissolution buffer.

For trypsin digestion, 3 μL of 1 μg/μL trypsin and 500 μL of 50 mM TEAB buffer were added. The sample was mixed and allowed to digest at 37 °C overnight. Following digestion, formic acid was added to the sample mixture, which was then centrifuged at 12,000 × $g$ for 5 min at room temperature. The resulting supernatant was gently loaded onto a C18 desalting column. The column was washed thrice with a washing buffer (0.1% formic acid, 3% acetonitrile) and then eluted three times with an elution buffer, which included elution buffer (0.1% formic acid, 70% acetonitrile). The eluted sample was collected and subsequently lyophilized.

LC-MS/MS analysis was carried out according to the methods described below. The lyophilized powder was dissolved in 10 μL of solution A (100% water, 0.1% formic acid), and then centrifuged at 14,000 × $g$ for 20 min at 4 °C. Subsequently, 1 μg of the sample was injected into a home-made C18 Nano-Trap column (2 cm × 75 μm, 3 μm). The peptides were separated using a home-made analytical column (15 cm × 150 μm, 1.9 μm) through a linear gradient elution. The separated peptides were analyzed using the Q Exactive HF-X mass spectrometer (Thermo Fisher). The ion source used was Nanospray Flex™ (ESI), with a spray voltage of 2.3 kV and an ion transport capillary temperature of 320 °C. The top 40 precursor ions with the highest abundance in the full scan were selected for fragmentation by higher energy collisional dissociation (HCD), and subsequently analyzed in MS/MS.

The output raw data were preceded to identify potential salivary proteins using the software MaxQuant 1.6.5.0 with the following default parameters, tolerance of one missed cleavages of trypsin, carbamidomethyl (C), oxidation (M), and Acetyl (Protein N-term). Finally, the identifications were filtered to 1% false discovery rate (FDR) at the peptide-spectrum match (PSM) level. To mitigate common contaminants, the Common Repository of Adventitious Proteins database was employed.

## Identification of HESPs

A total of nine RpHESPs were identified in the salivary samples of *R. pedestris*. To identify additional RpHESP homologs in *R. pedestris*, the RpHESPs were searched against the *R. pedestris* genome with a cut-off E-value of $10^{-10}$. No other RpHESPs were found. Furthermore, the nine RpHESPs were used as queries to search the NCBI nr database. The insect and bacterial HESPs homologues with a cut-off E-value of $10^{-10}$ were retrieved and used for subsequent analysis.

As there was limited information regarding insect HESP homologues in the NCBI nr database, transcriptomic datasets were retrieved from 60 species in the SRA repository (Supplementary Table 3). The raw data were de novo assembled using SPAdes (v3.13.0) with default parameters. Thereafter, the assembled contigs were searched against the RpHESPs to identify the potential HESP homologues in other insect species using BLASTX with a cut-off E-value of $10^{-10}$.

## Bioinformatic analysis of HESPs

The SignalP 5.0 Server (https://services.healthtech.dtu.dk/service.php?SignalP-5.0) was employed to predict the presence of signal peptides and cleavage sites. In addition, the Splign Server (https://www.ncbi.nlm.nih.gov/sutils/splign/splign.cgi) was utilized to predict genomic structure of HESP

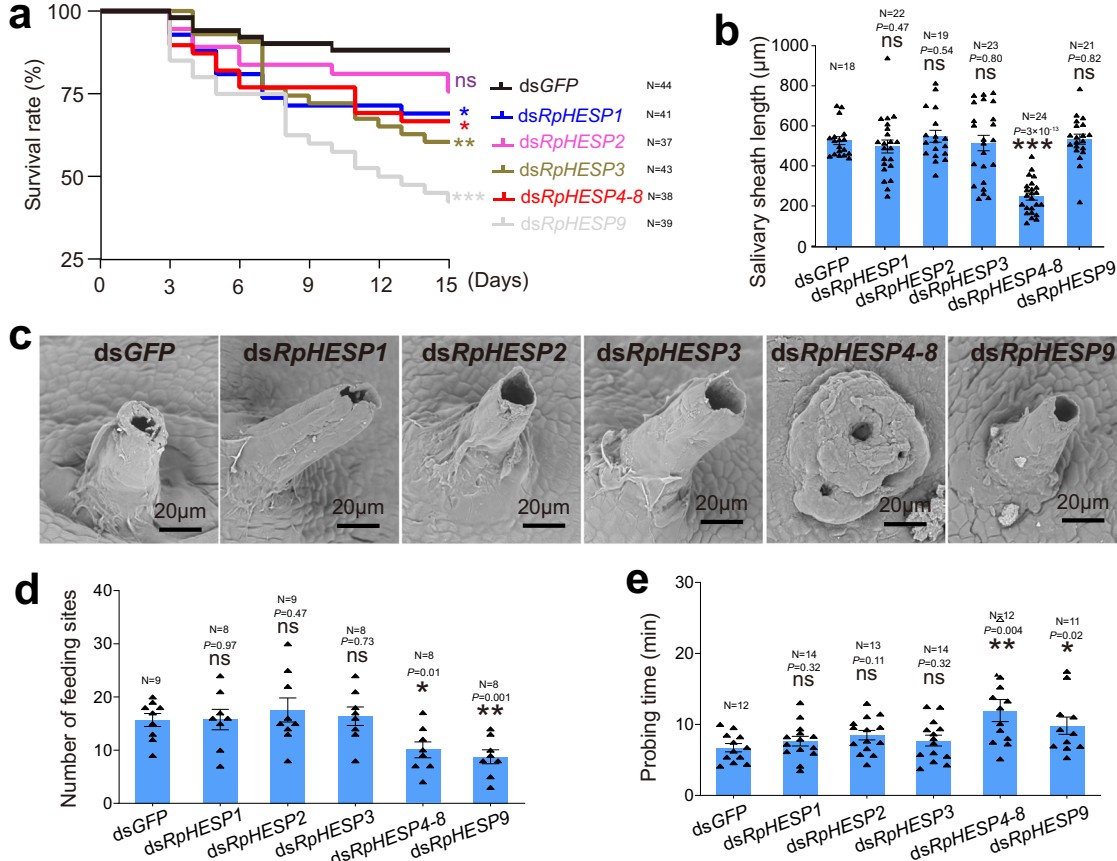

**Fig. 5 | Effects of ds*RpHESP* treatments on *Riptortus pedestris*. a** Survival rate after ds*RpHESP*s treatment. The third instar *R. pedestris* were injected with ds*RpHESP1*, ds*RpHESP2*, ds*RpHESP3*, ds*RpHESP4-8*, and ds*RpHESP9*, respectively, while individuals treated with ds*GFP* served as controls. The survival rates under each treatment were recorded for 15 consecutive days. *P*-values between ds*RpHESP* treatments and ds*GFP*-treated control were determined by log-rank test. **b** The length of salivary sheath secreted by dsRNA-treated *R. pedestris*. The salivary sheath left in soybean seeds were disserted and measured using ImageJ software. **c** Morphological observations of the salivary sheath after dsRNA treatment. The salivary sheath left on soybean seeds were observed under scanning electron microscopy. **d** Probing frequency of dsRNA-treated *R. pedestris*. The probing frequency was recorded by counting the number of salivary sheath left in soybean seeds. **e** Probing time of dsRNA-treated *R. pedestris*. The probing time was recorded by measuring the time from probing initiation to successful penetration. In **a**, **b**, **d**, and **e**, "N" represents the number of replicates used in each treatment. In **c**, more than 20 salivary sheaths were tested, and similar results were found. Data in **b**, **d**, and **e** were expressed as mean ± SEM. *P*-values between ds*RpHESP* treatments and ds*GFP*-treated control were determined by two-tailed unpaired Student's *t*-test. *$P < 0.05$; **$P < 0.01$; ***$P < 0.001$; ns, not significant.

genes, whereas MegAlign v7.1.0 was adopted to estimate the amino acid sequence similarity[36].

### Phylogenetic analysis

Currently, the genomes of three insects in Coreoidea, Pyrrhocoroidea, and Lygaeoidea, respectively, are available, including *R. pedestris*, *P. apterus*, and *O. fasciatus*[26,27]. In this study, the HESPs homologues in these species, together with bacterial genes were used for phylogenetic analysis. First of all, the amino acid sequences were aligned with MAFFT (v7.310) with default parameters (maxiterate:1000), and ambiguously aligned regions were trimmed by Gblock[37]. Secondly, the best-fit model of amino acid substitution (PROTGAMMAJTT) was evaluated by ModelTest-NG v0.1.6. Then, the maximum likelihood (ML) trees were constructed using RAxML v0.9.0 with 1000 bootstrap replications[38]. More details of all the reference sequences used in phylogenetic analysis are listed in Supplementary Table 6.

To analyze the species relationship, the coding regions of assembled contigs generated above were firstly predicted using TransDecoder v5.5.0. After filtering redundant alternative splicing events, the protein data set containing nonredundant transcripts was used to find the homologous pairs of sequences by using the all-versus-all BLASTp algorithm with a significant cutoff E-value of $10^{-5}$. Thereafter, the BLASTp result was converted into a normalized similarity matrix and processed using OrthoMCL v2.0.9 with

default parameters[39]. In addition, protein families were identified by Markov chain clustering MCL-14–137[40]. The phylogenetic tree was then constructed using single copy orthologues in each species (1:1:1 genes identified by OrthoMCL analysis), and *Drosophila melanogaster* was utilized to root the tree. Moreover, sequence alignment was performed by MAFFT v7.1.0. Conserved amino-acid sites were identified by TrimAl v1.2[41]. Furthermore, ModelTest-NG was employed to determine the best model (PROTGAM-MAJTT). Then, a ML tree was constructed using RAxML under the $LG + I + G4 + F$ model with 1,000 bootstraps replications. Bayesian MCMCTree v4.9i[42] was used to perform the divergence time analysis and calibration time was based on four nodes: *D. melanogaster-Bemisia tabaci* (330–376 Mya)[43], *Rhodnius prolixus- Halyomorpha halys* (108–227 Mya)[44], and *Diaphorina citri- Nilaparvata lugens* (112–290 Mya)[43].

### RNA interference (RNAi)

HESPs from *R. pedestris* and *P. apterus* were silenced using RNAi. To minimize the non-target effect of RNAi, the target genes were searched against the transcriptomic and genomic database of *R. pedestris* or *P. apterus*, and the specific regions of the target genes were selected for dsRNA synthesis. The DNA sequences of target genes were amplified using the primers listed in Supplementary Table 7, and later cloned into pClone007 Vector (Tsingke, Beijing, China). Meanwhile, the recombinant plasmids were amplified using

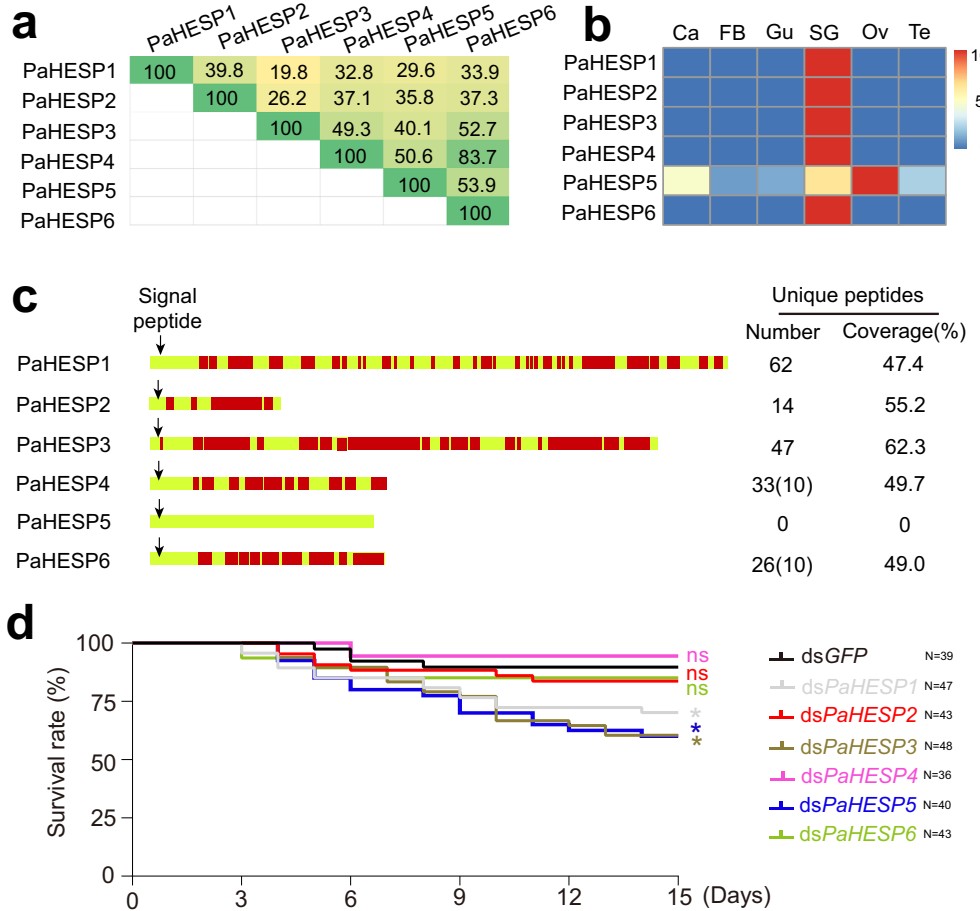

**Fig. 6 | Effects of ds*RaHESP* treatments on *Pyrrhocoris apterus*. a** Comparison of amino acid identity among PaHESPs. The pairwise distances between the amino acid sequences of six PaHESPs were calculated using MegAlign. **b** Expression patterns of PaHESPs in six tissues. Ca, carcass; FB, fat body, Gu, guts; SG, salivary glands; Ov, ovaries; Te, testis. The relative expression levels of PaHESPs were calculated based on qRT-PCR data and illustrated by a heat map. **c** Distribution of unique peptides identified by LC-MS/MS. Unique peptides mapped to PaHESPs are labeled in red. The number and coverage of unique peptides in each protein are displayed on the right. The number in the brackets indicates unique peptides belonging to more than one protein. Arrow indicates the signal peptide cleavage site. **d** The survival rate after ds*PaHESP*s treatment. The third instar *P. apterus* were injected with dsRNA, and the survival rates under each treatment were recorded for 15 consecutive days. "N" represents the number of replicates used in each treatment. *P*-values between ds*PaHESP* treatments and ds*GFP*-treated control were determined by log-rank test. *$P < 0.05$; ns, not significant.

the primers containing T7 sequences, and the PCR-generated DNA templates were adopted to synthesize the double-stranded RNAs (dsRNAs) using a T7 High Yield RNA Transcription Kit (Vazyme, Nanjing, China).

RNAi experiments were performed as previously described[45]. Briefly, the dsRNA was loaded into a capillary tube using tips. The *R. pedestris* and *P. apterus* that were pre-anaesthetized with carbon dioxide were placed on the plate constantly releasing carbon dioxide. Thereafter, dsRNA was injected into the insect mesothorax using FemtoJet (Eppendorf-Netheler-Hinz, Hamburg, Germany). Then, the treated insects were reared on soybean seeds or clover seeds, respectively. The silencing efficiency was determined at four days post-injection using the qRT-PCR.

### qRT-PCR

To determine the RNAi efficiency, the whole *R. pedestris* (4) and *P. apterus* (4) were used. To collect tissue samples, carcass (10), fat body (10), guts (20), and salivary glands (20) were carefully dissected from the 5th instar nymphs, whereas, testes (10) and ovaries (10) were dissected from male and female adults, respectively. The number of insects used in each sample was illustrated in brackets. Afterwards, total RNA was extracted using RNAiso plus (TaKaRa, Dalian, China). After determining the RNA integrity and quantity, the first-strand cDNA was reverse-transcribed using HiScript II Q RT SuperMix (Vazyme, Nanjing, China). Later, qPCR was run on a Roche Light Cycler® 480

Real-Time PCR System using the SYBR Green Supermix Kit (Yeasen, Shanghai, China). The following PCR conditions were used: denaturation for 5 min at 95 °C, followed by 40 cycles at 95 °C for 10 s and 60 °C for 30 s. Primer Premier v6.0 was used to design the qPCR primers (Supplementary Table 7). Actin and GAPDH were used as the house-keeping genes. The relative quantitative method ($2^{-\Delta\Delta Ct}$) was adopted to evaluate the relative gene expression[46]. Three independent biological replicates were performed.

### Scanning electron microscopy

The treated insects were allowed to feed on soybean seeds or clover seeds for 24 h. Scanning electron microscopy were performed as we previously described[6]. In detail, the infested seeds were attached to a stub and placed in a dryer at 65 °C for 2 h. Then, the seeds were further dried in a desiccator under vacuum condition. Gold-sputtering was performed when vacuum degree reached 6.5, and the gold-sputtering time was set as 15 s. Thereafter, the samples were visualized by SEM TM4000 II plus (Hitachi, Tokyo, Japan). More than 20 salivary sheaths in each treatment were observed.

### Insect bioassays

For the survival assay, the live insects at 24 h after dsRNA treatment were selected to exclude the insects died with mechanical injury. Thereafter, the insects were transferred to a glass tube (diameter, 10 mm; height, 20 cm)

containing soybean or clover seeds. The wet cotton was placed into two rearing devices, respectively, to provide the drinking water. Then, the survival rates under each treatment were recorded for 15 consecutive days. A group of 12 insects were included in each treatment, and three independent replicates were performed.

## Feeding behavior analysis

Before experiment, the dsRNA-treated *R. pedestris* (at 4-6 days post injection) were collected and fed on a filter paper for 24 h with water provided only. Subsequently, the insects were transferred into a glass tube with soybean seeds placed at the center. The probing time, which was recorded by measuring the time from probing initiation to successful penetration. The probing initiation was regarded as the first time the stylets contact the soybean seed. Successful penetration was regarded as the stylets anchored at a fixed point and inserted into seeds (the stylet was moveless once penetrated successfully). When the insect stylet contacts the soybean seed, we start the timer. When the stylet successfully penetrates into seeds, we stop the timer.

For analysis of probing frequency, a group of 4 dsRNA-treated *R. pedestris* were allowed to feed on soybean seeds for 48 h. The probing frequency was recorded by counting the salivary sheath left in soybean seeds under a SZ2-ILST Stereomicroscope, as *R. pedestris* secrete salivary sheath in each probing attempt.

In addition, the salivary sheath morphology was observed. The salivary sheath left in soybean seeds was carefully disserted under a SZ2-ILST Stereomicroscope using a pair of forceps. The disserted samples were carefully washed in 1× PBS to remove attached soybean components. Thereafter, the salivary sheath was photographed under a SMZ225 Stereomicroscope (Nikon, Tokyo, Japan). The salivary sheath length was measured using ImageJ software v1.53e (https://imagej.nih.gov/).

## Statistics and reproducibility

The log-rank test (SPSS Statistics 19, Chicago, IL, USA) was applied to determine the statistical significance of survival distributions ($n = 36–47$ biological replicates). Two-tailed unpaired Student's $t$ test was used to analyze the results of qRT-PCR ($n = 3$ biological replicates), salivary sheath length ($n = 18–24$ biological replicates), probing frequency ($n = 8–9$ biological replicates), and probing time ($n = 11–14$ biological replicates). The number of biological replicates and the exact $p$ value of each statistical test was provided in Supplementary Data 1. Data were graphed in GraphPad Prism 9.

## Reporting summary

Further information on research design is available in the Nature Portfolio Reporting Summary linked to this article.

## Data availability

The mass spectra data generated in this study were submitted to the ProteomeXchange with accession numbers PXD039150 and PXD039269. Sequences of PaHESPs and RpHESPs have been deposited in NCBI GenBank with accession numbers OQ126862-OQ126876. Source data are provided in Supplementary Data 1.

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

## Acknowledgements
We thank Hai-Jun Xu from Zhejiang University for kindly providing firebugs. This project has received funding from the National Natural Science Foundation of China (U20A2036: J.P.C., J.M.L.; 32001987: Y.H.L.), the Natural Science Foundation of Zhejiang Province (LDQ24C140001: H.J.H.), and the Young Elite Scientists Sponsorship Program by CAST (2021QNRC001: H.J.H.).

## Author contributions
H.-J.H., C.-X.Z., J.-M.L. and J.-P.C. planned and designed the research. H.-J.H., L.-L.L., Z.-X.Y., J.-B.L., Y.-H.L., and Z.-Y.W. performed experiments and analyzed data. L.-L.L. drew the model of HESP evolution. Z.-T.S. provided valuable suggestion for the research. H.-J.H. drafted the manuscript.

## Competing interests
The authors declare no competing interests.
