## [Peer Review File · Communications Biology]

Reviewers' comments:

Reviewer #1 (Remarks to the Author):

In the current manuscript (COMMSBIO-23-3422), Huang and co-authors investigated a new group of proteins that were identified from salivary secretions of *Riptortus pedestris* and *Pyrrhocoris apterus* (Hemiptera: Heteroptera). The authors claim that these genes were acquired by horizontal gene transfer (HGT) from bacterial donor species. Transcription of this gene family is restricted to the salivary glands where high mRNA levels were observed. Interestingly, one gene in *P. apterus* exhibited an aberrant spatial transcriptional pattern (PaESSP5), suggestive of other biological functions. Using reverse genetics (RNAi), the functional importance of this gene family is shown for insect survival and feeding on soybean. By knocking down transcription, RpESSP4-8 appears critical for the formation of the salivary sheath in *R. pedestris*. The manuscript is generally well-written and the M&M mostly explained in sufficient detail. Although the role of HGT for insect herbivores is well established, the authors uncover a potential new function of laterally acquired genes (see below). The manuscript currently suffers from several significant issues (see below, major1-3). However, I believe that a very thorough/extensive major revision could address these concerns. If these revisions continue to support the current conclusions, this study has the potential to benefit the broad field of insect physiology and herbivory.

MAJOR1. The current analyses are insufficient and too shallow to support an HGT scenario. The authors now adopt two approaches to conclude that RSSP's originated by HGT; (1) a phylogenetic reconstruction and (2) an -omic screen.

(1) A phylogenetic reconstruction only suggests an HGT when the focal genes are clustered with homologs of non-related organisms by strongly supported branches. Unfortunately, the current tree (figure 3) thus does not support an HGT event. I would advise the authors to include a more exhaustive set of closely and step-wise more distantly related homologous proteins. Why did the authors restrict themselves to only including insect and bacterial "RSSP"? Fungal genomes do not code for RSSP genes? These could be added as important outgroups. In line with these suggestions, please root the phylogenetic tree in order to show the correct relatedness.

(2) The authors also screened a transcriptomic database to map the phylogenetic distribution of RSSP's. I was surprised to read that the initial screen was based on the raw short Illumina reads and not on assembled transcriptomes. How can the authors guarantee that divergent homologs were not missed? Indeed, such approaches fail to uncover more distant homology. Further, transcriptome assemblies are typically riddled with contaminants of fungal and bacterial species (at least, in my personal experience). I would strongly advise to first filter these assemblies to remove contaminating contigs.

(3) As the localization of the candidate HGT genes on the nuclear genome is key to support a HGT scenario, I feel that the genomic loci should be characterized and discussed in greater detail. Importantly, the short length of the scaffolds worries me. How can the authors be sure that these are not contaminant scaffolds? Are the neighbouring genes typical eukaryotic genes? Are there transposable elements nearby? Notably, to test that the genome does not suffer from misassembly, the authors could perform PCR reactions to proof that the HGT genes are physically linked to its predicted genomic locus.

MAJOR2. The authors did not investigate potential off-target effects of RNAi silencing. As the authors want to link specific paralogs to specific functions, this is absolutely crucial. How can the authors be sure that exposing the insects to eg dsRpESSP1, does not affect the transcription of the other ESSP paralogs? This is especially pertinent because some of the paralogs exhibit near-identical coding sequences (as far as I understand). Quantifying (potential) off-target effects of certain dsRNA molecules on a multi-gene family is common practice.

MAJOR3. The introduction and discussion lacks a lot of critical references and therefore depth. The role of HGT on insect plant feeding is well established. See for instance <https://doi.org/10.1073/pnas.2205857119> and <https://doi.org/10.1093/gbe/evw119> . This rich literature does not take away the novelty of the current study as a potentially new function is uncovered here. I would strongly recommend the authors to (briefly) introduce this long research history and to discuss their findings within the existing body of work.

More minor issues:

Title - Personally, I feel that the title is a bit too vague and does not highlight the novelty well enough.

Line 12 – Perhaps not the best idea to already drop ‘orphan’ in the first sentence.

Line 52 – I would remove ‘accidental’ here and elsewhere.

Line 73 and elsewhere – I am not a fan of the term ESSP because I think it is confusing – are all ESSPs horizontally transferred? I would guess that there are also vertically acquired orphans within Eutrichophora. I would use the putative (ancestral) function to name these enzymes.

Line 106 – I am not an expert on Hemiptera taxonomy but I would be really surprised that there are no references for hemipteran phylogeny?

Line 192 until 194 – this sentence feels out of place, I would remove.

Line 206-207 – a more in-depth phylogenetic analysis would address these issues!

Line 222 – what are ‘forward roles’?

Figures

Figure 4 and 5 – I would show confidence intervals and add the number of replicates.

Figure 6, I would remove this figure and place a detailed figure on the genomic loci (see MAJOR comment) in the main text.

Line 245 – 247 – this section lacks detail – How were the sheaths and affected areas collected?

Reviewer #2 (Remarks to the Author):

Huang et al. presented an investigation on the salivary proteins of herbivorous insects originated from horizontal gene transfer. The authors found that ESSPs in insects were originated from bacteria and underwent horizontal gene transfer. And the function of ESSPs in different insects were tested.

This study makes an important contribution and lays the foundation to potential origin of orphan genes in insect saliva and shed light on the evolution of horizontally transferred genes. I still have some comments. It would be better if the authors improve the methods or explanation of the methods in detail. My comments are as follows.

1. The estimation of divergence time is not accepted. The analysis of molecular clock is quite common in all entomological study. The birth-death on divergence time are always applied in BEAST, because it can easily indicate the time. I would not suggest use TIMETREE to predict the prior node in the phylogenetic analysis.

2. How did author conclude 78-95 million years ago based on the phylogeny tree?

3. Lines 143-149, why did authors believe ESSPs play a role in probing, but not other feeding behaviors? EPG can be more accurate to show the process of penetration and salivation. More data of feeding behaviors are required. In addition, the condition and stage of plant are also important. Fresh tissue always attracts more bugs for eating. Please provide more information about the plant (condition, feeding locations [ideally, bugs were only allowed to stay in the similar location of plant to start feeding]) in Methods section. Besides, biological replicates should be indicated in these tests.

4. Line 82, no RpESSPs were found in *R. pedestris* genome, why authors concluded that *R. pedestris* encode nine RpESSPs?

5. Line 157 and Fig. 4c, it is not notable that the tubular structure was collapsed in salivary sheaths, and the shooting direction was different from the control.

6. Biological replicates should be shown in the method "collection of salivary secretions".

Reviewer #3 (Remarks to the Author):

Herbivorous insects employ an array of orphan salivary proteins to facilitate feeding. However, the mechanisms behind the evolution of these orphan genes remain poorly understood. In this study, the authors provide evidence for the horizontal transfer of a bacterial gene to the ancestral of Coreoidea, Lygaeoidea, and Pyrrhocoroidea. The acquired ESSP genes undergo duplications, gain introns, and evolve in recipient insects and serve as the main saliva components that enhance insect feeding. Silencing of several ESSPs increases the difficulty of *R. pedestris* in probing the soybean, and the treated insects display a decreased survivability. Overall, this study is well performed, innovative and appropriate techniques were used for all experiments. The data are robust and generally support the author's conclusions, and the paper is well written. I have a couple of suggestions that the authors could address before publication of this manuscript.

- 1) The authors found that silencing of RpESSP4-8 collectively affects the probing and salivary sheath formation of *R. pedestris*. Since gene silencing is very effective in *R. pedestris*, why don't you silence RpESSP4, RpESSP5, RpESSP6, RpESSP7 and RpESSP8 separately? By this way, you can know which ESSP is responsible for probing and salivary sheath formation.
- 2) How do RpESSP4-8 involve in salivary sheath formation? Are these proteins directly form the salivary sheath? To answer this question, I would suggest the author to do a LC-MS/MS to analyze the components of the salivary sheath.
- 3) Material and methods, 11. Feeding behavior recording. How did you measure the time from probing initiation to successful penetration? How many replicates and how many insects in each replicate? Please provide more information. Why don't you use EPG to analyze the feeding behavior of the insect?
- 4) Material and methods, 9. Scanning electron microscopy (SEM). The description of SEM is too simple. How many replicates have you done for each group? Please provide more details.

Reviewer 1

In the current manuscript (COMMSBIO-23-3422), Huang and co-authors investigated a new group of proteins that were identified from salivary secretions of *Riptortus pedestris* and *Pyrrhocoris apterus* (Hemiptera: Heteroptera). The authors claim that these genes were acquired by horizontal gene transfer (HGT) from bacterial donor species. Transcription of this gene family is restricted to the salivary glands where high mRNA levels were observed. Interestingly, one gene in *P. apterus* exhibited an aberrant spatial transcriptional pattern (PaHESP5), suggestive of other biological functions. Using reverse genetics (RNAi), the functional importance of this gene family is shown for insect survival and feeding on soybean. By knocking down transcription, RpHESP4-8 appears critical for the formation of the salivary sheath in *R. pedestris*. The manuscript is generally well-written and the M&M mostly explained in sufficient detail. Although the role of HGT for insect herbivores is well established, the authors uncover a potential new function of laterally acquired genes (see below). The manuscript currently suffers from several significant issues (see below, major1-3). However, I believe that a very thorough/extensive major revision could address these concerns. If these revisions continue to support the current conclusions, this study has the potential to benefit the broad field of insect physiology and herbivory.

Response: We greatly appreciate your valuable comments and constructive suggestion to our work. We have carefully revised the MS following your valuable suggestions and comments.

MAJOR1. The current analyses are insufficient and too shallow to support an HGT scenario. The authors now adopt two approaches to conclude that RSSP's originated by HGT; (1) a phylogenetic reconstruction and (2) an -omic screen.

(1) A phylogenetic reconstruction only suggests an HGT when the focal genes are clustered with homologs of non-related organisms by strongly supported branches. Unfortunately, the current tree (figure 3) thus does not support an HGT event. I would advise the authors to include a more exhaustive set of closely and step-wise more distantly related homologous proteins. Why did the authors restrict themselves to only including insect and bacterial "RSSP"? Fungal genomes do not code for RSSP genes? These could be added as important outgroups. In line with these suggestions, please root the phylogenetic tree in order to show the correct relatedness.

Response: Thank you for your suggestion. There is no HESP homology in fungi or other protists, except for bacteria. Therefore, it is difficult to infer which species should be placed as an outgroup. In the revised version, we use unrooted tree for the phylogenetic analysis.

All sequences associated with HESP have been used for phylogenetic construction. Given the limited sequences available for HESPs, the evidence for HGT is not very robust. Therefore, in the manuscript, we tone down the description of HGT, and discussed that "In this study, HESPs was confirmed to be integrated into insect genome (Figure 2; Table S2), strictly distributed in three insect superfamily (Figure 3; Table S3), and showed high sequence similarity to bacteria genes (Figure 4), suggesting a potential HGT origin. However, compared with most reported HGT events that had a lot of homologues in donor relatives, only a few bacteria species contain HESP homology. Moreover, in current phylogenic tree, insect HESPs did not cluster with bacterial HESPs in strongly supported bootstrap (Figure 4). Based on above results, our study only provided a possible origin of HESP. The exact phylogeny of HESP still needs further investigations. Perhaps a high-throughput sequencing of a wider range of organisms in the future might solve this issue." in Line 235-243.

(2) The authors also screened a transcriptomic database to map the phylogenetic distribution of RSSP's. I was surprised to read that the initial screen was based on the raw short Illumina reads and not on assembled transcriptomes. How can the authors guarantee that divergent homologs were not missed? Indeed, such approaches fail to uncover more distant homology. Further, transcriptome assemblies are typically riddled with contaminants of fungal and bacterial species (at least, in my personal experience). I would strongly advise to first filter these assemblies to remove contaminating contigs.

Response: Thank you for your suggestion. In the revised version, the transcriptomic data that did not identify HESP homology were also assembled. By blast search, we confirmed that no HESP homology was identified in these species, and HESP was restricted to Eutrichophora species.

For the exclusion of bacterial contamination, the flanking sequences of the HESP in each contig were searched for potential bacterial origin. The results showed that these flanking sequences were not belong to bacteria. Additionally, given that flanking genes of HESP in the genome of *Riptortus pedestris*, *Oncopeltus fasciatus* and *Pyrrhocoris apterus* were belong to insect. It is reliable that HESP sequences in Eutrichophora species were belonged to insect, but not due to bacterial contamination.

(3) As the localization of the candidate HGT genes on the nuclear genome is key to support a HGT scenario, I feel that the genomic loci should be characterized and discussed in greater detail. Importantly, the short length of the scaffolds worries me. How can the authors be sure that these are not contaminant scaffolds? Are the neighbouring genes typical eukaryotic genes? Are there transposable elements nearby? Notably, to test that the genome does not suffer from misassembly, the authors could perform PCR reactions to proof that the HGT genes are physically linked to its predicted genomic locus.

Response: Thank you for your suggestion. The *Riptortus pedestris* genome was assembled at a chromosomal level, while the *Pyrrhocoris apterus* and *Oncopeltus fasciatus* genomes were assembled at a scaffold-level. In the revised version, the flanking regions of HESPs in the genome were analyzed, and five upstream and downstream genes adjacent to HESPs were displayed in Table S2. The results showed that the flanking genes of HESPs were belonged to insects, indicating that the HESP-contained scaffolds/chromosomes were belong to insect, but not the contaminants. Additionally, gene structure analysis demonstrated that all HESPs in three Eutrichophora species display exon-intron structure, which further validate that these HESPs were belong to insect, but not the prokaryote. For transposable elements, we predicated a Gypsy LTR retrotransposon at the *R. pedestris* intergenic region between RpHESP8 and RpHESP9.

Additionally, the sequence between RpHESP1 and *R. pedestris* bifunctional purine biosynthesis (validated to be insect origin) was amplified by PCR. The result showed that RpHESP1 was physically linked to insect gene, and the genome assembly was reliable.

MAJOR2. The authors did not investigate potential off-target effects of RNAi silencing. As the authors want to link specific paralogs to specific functions, this is absolutely crucial. How can the authors be sure that exposing the insects to eg dsRpHESP1, does not affect the transcription of the other HESP paralogs? This is especially pertinent because some of the paralogs exhibit near-identical coding sequences (as far as I understand). Quantifying (potential) off-target effects of certain dsRNA molecules on a multi-gene family is common practice.

Response: Thank you for your suggestion. To avoid off-target effects, the dsRNA was designed in region where no nucleic acid similarity was detected in other paralogs. We addressed this in the revised version. Additionally, the effect of dsRNA treatment on all other paralogs were investigated in the revised version. The results showed that no off-target effects were found in *Riptortus pedestris* or *Pyrrhocoris apterus*, and the dsRNA specifically and effectively reduced the expression of the target genes, respectively. In the revised version, we detailed described these results as follow: “First, dsRNA targeting RpHESP1, RpHESP2, RpHESP3, RpHESP4-8, and RpHESP9 were individually injected into third instar nymphs. Four days after the injection, the RNAi efficacy was evaluated, and the results showed that the dsRNA specifically and effectively reduced the expression of the target genes, respectively (Figure S4). It is noteworthy that the expression of *RpHESP4-8* and *RpHESP9* were significantly induced when knocked down RpHESP1 or RpHESP2, indicating the potential compensatory effect”, “The dsRNA targeting PaHESP1, PaHESP2, PaHESP3, PaHESP4, PaHESP5, and PaHESP6 were individually injected into third instar nymphs, respectively. qRT-PCR results demonstrated that the dsRNA can specifically and effectively reduce the expression of the target genes, respectively (Figure S6)”.

MAJOR3. The introduction and discussion lacks a lot of critical references and therefore depth. The role of HGT on insect plant feeding is well established. See for instance <https://doi.org/10.1073/pnas.2205857119> and <https://doi.org/10.1093/gbe/evw119> . This rich literature does not take away the novelty of the current study as a potentially new function is uncovered here. I would strongly recommend the authors to (briefly) introduce this long research history and to discuss their findings within the existing body of work.

Response: Thank you for your suggestion. In the revised version, we introduce the importance of HGT in plant-insect interaction as follow: “HGT also plays critical roles in shaping the plant-insect interaction. Accumulated evidence demonstrated that HGT from microbial species enhanced the enzymatic repertoire of herbivorous insects, which in turn facilitated their adaptation toward herbivory and novel host plants. For example, the widespread of horizontally transferred polygalacturonases, cellulases, and xylanases in insect genome provided selective advantage to recipients by degrading plant cell wall components, while the β -fructofuranosidase, cysteine synthase, and biotin synthase involved in assimilation of plant nutrients. Additionally, insects have evolved various counter adaptations to plant defenses by acquiring pre-evolved genes from bacteria and fungi. In *Bemisia tabaci*, a fungal transferred salivary protein was reported to be migrated into plant cells and promoted insect feeding by suppressing ferredoxin-triggered plant immunity. Generally, the majority of reported horizontally transferred genes that were associated with insect feeding code for enzymes, and function similarly between the donor and recipient organisms. Our knowledge on their evolved function after co-opted by the insect recipient is still limited” in line 56-69. Also, we discussed our findings with these previous works.

More minor issues:

Title - Personally, I feel that the title is a bit too vague and does not highlight the novelty well enough.

Response: Thank you for your suggestion. In the revised version, the title was changed as “salivary proteins that potentially derived from horizontal gene transfer are critical for insect salivary sheath formation and other feeding process”

Line 12 – Perhaps not the best idea to already drop ‘orphan’ in the first sentence.

Response: Done. In the revised version, we avoid to use “orphan” in the abstract. In the introduction section, we briefly introduce the definition of “orphan genes” after its first appearance.

Line 52 – I would remove ‘accidental’ here and elsewhere.

Response: Done. We move it throughout the manuscript.

Line 73 and elsewhere – I am not a fan of the term HESP because I think it is confusing – are all HESPs horizontally transferred? I would guess that there are also vertically acquired orphans within Eutrichophora. I would use the putative (ancestral) function to name these enzymes.

Response: Sorry for the confusion. This gene was primary annotated as “uncharacterized protein”, we cannot infer its ancestral function. Given that the majority of genes in this family are salivary protein, and they are specifically distributed in Eutrichophora, we annotated it as “horizontal-transferred, Eutrichophora-specific salivary protein (HESPs)”, which distinguish it from other salivary proteins or Eutrichophora-specific proteins.

Line 106 – I am not an expert on Hemiptera taxonomy but I would be really surprised that there are no references for hemipteran phylogeny?

Response: Sorry for the confusion. This result was derived from the hemipteran phylogeny constructed in this study. In the revised version, we rewrote this section as “An evolutionary analysis based on 34 single copy genes demonstrated that Eutrichophora species (Coreoidea, Lygaeoidea, and Pyrrhocoroidea superfamilies) clustered in the same clade and that Eutrichophora diverged from the Pentatomoidea superfamily at approximately 78-95 million years ago (Mya) (Figure 3). Given that HESP distributed in nearly all Eutrichophora species, but not the Pentatomoidea species, we speculated that HESP might be evolved approximately 78-95 Mya, the time during the divergency of Eutrichophora and Pentatomoidea” in line 120-126.

Line 192 until 194 – this sentence feels out of place, I would remove.

Response: Thank you for your suggestion. We move it in the revised version.

Line 206-207 – a more in-depth phylogenetic analysis would address these issues!

Response: Thank you for your suggestion. In the revised version, we discussed this issue with phylogenetic analysis as follow: “In contrast, although all RpHESPs in *R. pedestris* located in the same chromosome, RpHESP1 was approximately 1.5×10^7 -bp far away from the other RpHESPs and had different transcription orientations (Figure 2a). Phylogeny analysis demonstrated that RpHESP1 was clustered with PaHESP1 and OfHESP1, but not the other RpHESPs (Figure 4). Therefore, we cannot rule out the possibility that RpHESP1 and the other RpHESPs derived from two independent HGT events, which deserved further investigations” in Line 251-257.

Line 222 – what are ‘forward roles’?

Response: Sorry for the confusion. We use “critical roles” in the revised version.

Figures

Figure 4 and 5 – I would show confidence intervals and add the number of replicates.

Response: Done. In the revised version, the dot chart was used to display the results, with each replicate represented by one dot. The exact p-value and the number of replicates were provided in the figure. Moreover, all data generated in this study were provided in Source Data File.

Figure 6, I would remove this figure and place a detailed figure on the genomic loci (see MAJOR comment) in the main text.

Response: Done. Thank you for your suggestion. This figure was moved to supplementary file. The genomic loci figure was placed in the main text.

Line 245 – 247 – this section lacks detail – How were the sheaths and affected areas collected?

Response: Done. Sorry about inadequate description. In the revised version, we provided the detail as follows: “The presence of a salivary sheath indicated the feeding site of the insects, which can be easily observed under a SZ2-ILST Stereomicroscope (Olympus, Shinjuku, Japan). To analyze the component of entire salivary secretion, the salivary sheath and the surrounding area near the salivary sheath were carefully collected using a pair of forceps (Ideal-Tek, Balerna, Switzerland). To analyze the component of salivary sheath only, the tubular structure outside the soybean seeds were cut, and rinsed in $1 \times$ PBS to remove potential watery saliva remained in the salivary sheath” in Line 298-306.

Reviewer 2

Huang et al. presented an investigation on the salivary proteins of herbivorous insects originated from horizontal gene transfer. The authors found that HESPs in insects were originated from bacteria and underwent horizontal gene transfer. And the function of HESPs in different insects were tested. This study makes an important contribution and lays the foundation to potential origin of orphan genes in insect saliva and shed light on the evolution of horizontally transferred genes. I still have some comments. It would be better if the authors improve the methods or explanation of the methods in detail. My comments are as follows.

Response: Thank you very much for your positive comments.

1. The estimation of divergence time is not accepted. The analysis of molecular clock is quite common in all entomological study. The birth-death on divergence time are always applied in BEAST, because it can easily indicate the time. I would not suggest use TIMETREE to predict the prior node in the phylogenetic analysis.

Response: Thank you for pointing out this. In the revised version, we use Bayesian MCMCTree to analyze divergence time as previous described (Johnson *et al.*, Phylogenomics and the evolution of hemipteroid insects, *PNAS*, 2018). The result showed that Eutrichophora diverged from the Pentatomoidea superfamily at approximately 117-141 mya, which is close to estimated time of that paper (Johnson *et al.*, 2018).

2. How did author conclude 78-95 million years ago based on the phylogeny tree?

Response: Sorry for the confusion. In the revised version, we detailly explained this conclusion as “Given that HESP distributed in nearly all Eutrichophora species, but not the Pentatomoidea species (Table S3), we speculated that HESP might be evolved approximately 117-141 Mya, the time during the divergency of Eutrichophora and Pentatomoiden” in Line 124-127.

3. Lines 143-149, why did authors believe HESPs play a role in probing, but not other feeding behaviors? EPG can be more accurate to show the process of penetration and salivation. More data of feeding behaviors are required. In addition, the condition and stage of plant are also important. Fresh tissue always attracts more bugs for eating. Please provide more information about the plant (condition, feeding locations [ideally, bugs were only allowed to stay in the similar location of plant to start feeding]) in Methods section. Besides, biological replicates should be indicated in these tests.

Response: Thank you for pointing out this. Both *R. pedestris* and *P. apterus* depend on seeds for completing life cycle. Therefore, we only use soybean seeds and clover seeds, but not the plant throughout the experiment. When using soybean seeds as food source, we cannot perform EPG analysis due to the technical difficulty (the plant electrode of EPG cannot insert into tiny seed, and no current loop formed). Therefore, we use other three parameters to infer the insect feeding process.

1) Previous work demonstrated that salivary sheath is critical for arching insect stylet at a fixed point. Insect (i.e., planthopper and aphid) that deficient in salivary sheath formation failed or difficult to start penetration. Therefore, the probing time, which was recorded by measuring the time from probing initiation to successful penetration, was compared. The results showed that the *dsRpHESP4-8-* and *dsRpHESP9-*treated insects repeatedly attempted to probe the soybean but difficult to start penetration.

2) The insect probing frequency was also critical for investigating insect feeding. We compared insect probing frequency by counting the salivary sheath left in soybean seeds, as *R. pedestris* secrete salivary sheath in each probing attempt. The results showed that the *dsRpHESP4-8-* and *dsRpHESP9-*treated *R. pedestris* probed less frequently than that of the *dsGFP* control.

3) Salivary sheath morphology is also an important indicator for insect feeding activity, as it always envelops stylets and is left in plant tissues after stylet withdrew. We dissected salivary sheath left in soybean seeds, and compared their morphology. The results showed that the salivary sheath left in soybean seeds was very short in *dsRpHESP4-8-*treated *R. pedestris*, suggesting that the insect stylet cannot be not deeply inserted into food source.

For the biological replicates, in the revised version, the dot chart was used to displayed the results, with each replicate represented by one dot. The exact p-value and the number of replicates were provided in the figure or figure legends. Moreover, all data generated in this study were provided in Source Data File.

4. Line 82, no RpHESPs were found in *R. pedestris* genome, why authors concluded that *R. pedestris* encode nine RpHESPs?

Response: Sorry for the confusion. In the revised version, we rewrote this sentence as “To identify additional RpHESP homologs in *R. pedestris*, the nine RpHESPs were searched against the *R. pedestris* genome. In addition to these nine RpHESPs, we did not find other RpHESPs. These results suggest that all RpHESPs in *R. pedestris* are secreted into plants during insect feeding” in Line 89-92.

5. Line 157 and Fig. 4c, it is not notable that the tubular structure was collapsed in salivary sheaths, and the shooting direction was different from the control.

Response: Thank you for pointing out this. All salivary sheath secreted by *dsRpHESP4-8-*treated *R. pedestris* were significantly different from that of the control. We observe more than 20 salivary sheaths, and similar results were found. In the revised version, we displayed figures from the same shooting direction in Fig. 5c.

6. Biological replicates should be shown in the method "collection of salivary secretions".

Response: Done.

Reviewer 3

Herbivorous insects employ an array of orphan salivary proteins to facilitate feeding. However, the mechanisms behind the evolution of these orphan genes remain poorly understood. In this study, the authors provide evidence for the horizontal transfer of a bacterial gene to the ancestral of Coreoidea, Lygaeoidea, and Pyrrhocoroidea. The acquired HESP genes undergo duplications, gain introns, and evolve in recipient insects and serve as the main saliva components that enhance insect feeding. Silencing of several HESPs increases the difficulty of *R. pedestris* in probing the soybean, and the treated insects display a decreased survivability. Overall, this study is well performed, innovative and appropriate techniques were used for all experiments. The data are robust and generally support the author's conclusions, and the paper is well written. I have a couple of suggestions that the authors could address before publication of this manuscript.

Response: Thank you very much for your positive comments.

1) The authors found that silencing of RpHESP4-8 collectively affects the probing and salivary sheath formation of *R. pedestris*. Since gene silencing is very effective in *R. pedestris*, why don't you silence RpHESP4, RpHESP5, RpHESP6, RpHESP7 and RpHESP8 separately? By this way, you can know which HESP is responsible for probing and salivary sheath formation.

Response: Thank you for pointing out this, and sorry for the confusion. RpHESP4-8 shared >92% nuclear acid sequence similarity, which cannot be specifically silenced by RNAi. In the revised version, we displayed the sequence alignment results of RpHESP4-8 in Supplementary Figure 3, and addressed that "As RpHESP4 to RpHESP8 shared >92% nuclear acid sequence similarity, and it is difficult to silence these genes independently. Therefore, RpHESP4 to RpHESP8 were collectively silenced using the conserved region (refer to RpHESP4-8)."

2) How do RpHESP4-8 involve in salivary sheath formation? Are these proteins directly form the salivary sheath? To answer this question, I would suggest the author to do a LC-MS/MS to analyze the components of the salivary sheath.

Response: Thank you for your suggestion. In the revised version, the salivary sheath component was further analyzed using LC-MS/MS analysis. The tubular structure outside the soybean seeds were cut and washed to remove potential watery saliva. The results showed that all RpHESPs can be abundantly identified in salivary sheath. These results suggest that all RpHESPs are present in salivary sheath, but only RpHESP4-8 involves in maintaining the structure of the salivary sheath.

3) Material and methods, 11. Feeding behavior recording. How did you measure the time from probing initiation to successful penetration? How many replicates and how many insects in each replicate? Please provide more information. Why don't you use EPG to analyze the feeding behavior of the insect?

Response: Thank you for pointing out this. In the revised version, we addressed that "The probing time, which was recorded by measuring the time from probing initiation to successful penetration. The probing initiation was regarded as the first time the stylets contact the soybean seed. Successful penetration was regarded as the stylets anchored at a fixed point and inserted into seeds (the stylet was moveless once penetrated successfully). When the insect stylet contacts the soybean seed, we start the timer. When the stylet successfully penetrates into seeds, we stop the timer. More than 12

insects were tested” in Line 443-448.

For the information of replicates, we provide them in method section, in figure legends in detail. The dot chart was used to displayed the results, with each replicate represented by one dot. The exact p-value was provided in the figure. Moreover, all data generated in this study were provided in Source Data File.

In this study, we did not use EPG technology due to the technical difficulty (the plant electrode of EPG cannot insert into tiny seed, and no current loop formed). Therefore, we use other three parameters to infer the insect feeding process: 1) Previous work demonstrated that salivary sheath is critical for arching insect stylet at a fixed point. Insect (i.e., planthopper and aphid) that deficient in salivary sheath formation failed or difficult to start penetration. Therefore, the probing time, which was recorded by measuring the time from probing initiation to successful penetration, was compared. The results showed that the *dsRpHESP4-8-* and *dsRpHESP9-*treated insects repeatedly attempted to probe the soybean but difficult to start penetration. 2) The insect probing frequency was also critical for investigating insect feeding. We compared insect probing frequency by counting the salivary sheath left in soybean seeds, as *R. pedestris* secrete salivary sheath in each probing attempt. The results showed that the *dsRpHESP4-8-* and *dsRpHESP9-*treated *R. pedestris* probed less frequently than that of the *dsGFP* control. 3) Salivary sheath morphology is also an important indicator for insect feeding activity, as it always envelops stylets and is left in plant tissues after stylet withdrew. We dissected salivary sheath left in soybean seeds, and compared their morphology. The results showed that the salivary sheath left in soybean seeds was very short in *dsRpHESP4-8-*treated *R. pedestris*, suggesting that the insect stylet cannot be not deeply inserted into food source.

4) Material and methods, 9. Scanning electron microscopy (SEM). The description of SEM is too simple. How many replicates have you done for each group? Please provide more details.

Response: Done. In the revised version, we provide the detail procedure of SEM, and addressed the replicates in methods and figure legends as follow: “The treated insects were allowed to feed on soybean seeds or clover seeds for 24 h. Scanning electron microscopy were performed as we previously described 6. In detail, the infested seeds were attached to a stub and placed in a dryer at 65°C for 2 h. Then, the seeds were further dried in a desiccator under vacuum condition. Gold-sputtering was performed when vacuum degree reached 6.5, and the gold-sputtering time was set as 15s. Thereafter, the samples were visualized by SEM TM4000 II plus (Hitachi, Tokyo, Japan). More than 20 salivary sheaths in each treatment were observed” in Line 442-428.

Reviewers' comments:

Reviewer #1 (Remarks to the Author):

I would like to thank the authors for their thorough revisions. I feel that the general quality of the manuscript greatly improved, especially related to the genomic position of the HESP genes. However, the revised phylogenetic and -omic mining results are worrisome. In their rebuttal, the authors are correct to strongly nuance their initial conclusion that the HESP genes were acquired by HGT. As a result, the revisions no longer fully support one of the major conclusions of the manuscript (ie that the genes originated via HGT) that now largely remains a running hypothesis. It remains perfectly possible now that the HESP genes were in fact vertically transmitted orphan genes. Indeed, based on the new results, how can the authors rule out convergent molecular evolution to explain the current phylogenetic reconstruction (figure 4 of revised manuscript). Also, how can the authors rule out that the HGT event was not from insects into bacteria? I realize that the authors are constrained by the limited genomic/genic public resources to address these unresolved issues. For future research to identify potential bacterial donors and to solidify the claim of HGT, perhaps the authors could consider sequencing the genomes of associated symbionts. But for the current manuscript, I regret to conclude that the revisions did not offer the necessary solidity and feel that the manuscript cannot be accepted for publication in its present form.

Reviewer #2 (Remarks to the Author):

This resubmitted manuscript is carefully revised and well presented. The revised discussion and methods are up to date, the analyses in the result is adequate. It seems much better than the last one. But I still have some minor comments.

- 1) Authors focused on *Riptortus pedestris* and *Pyrrhocoris apterus* in most study, but only *Oncopeltus fasciatus* phylogenetical analysis was shown. The reason in the introduction (line75) appeared to be farfetched, because many important species are in those three superfamilies. I suggest authors explain the reasons in the Method section through referring to the phylogenetic relationship and useful genomic data of *Oncopeltus fasciatus*.
- 2) Even author updated and used the new name HESP in the whole manuscript, there are still some ESSP in figure 1, figure 2 and figure 6. Please change all ESSP to HESP in the figures.
- 3) Line 638 should be "Effects of dsPaHESP treatments on *Pyrrhocoris apterus*".
- 4) Figure 3, the name of bacteria protein should not be HESP. Because authors defined HESP as horizontal-transferred, *Eutrichophora*-specific salivary protein.

Reviewer #3 (Remarks to the Author):

The authors have addressed all of my concerns.

Reviewer #1 (Remarks to the Author):

I would like to thank the authors for their thorough revisions. I feel that the general quality of the manuscript greatly improved, especially related to the genomic position of the HESP genes. However, the revised phylogenetic and -omic mining results are worrisome. In their rebuttal, the authors are correct to strongly nuance their initial conclusion that the HESP genes were acquired by HGT. As a result, the revisions no longer fully support one of the major conclusions of the manuscript (ie that the genes originated via HGT) that now largely remains a running hypothesis. It remains perfectly possible now that the HESP genes were in fact vertically transmitted orphan genes. Indeed, based on the new results, how can the authors rule out convergent molecular evolution to explain the current phylogenetic reconstruction (figure 4 of revised manuscript). Also, how can the authors rule out that the HGT event was not from insects into bacteria? I realize that the authors are constrained by the limited genomic/genic public resources to address these unresolved issues. For future research to identify potential bacterial donors and to solidify the claim of HGT, perhaps the authors could consider sequencing the genomes of associated symbionts. But for the current manuscript, I regret to conclude that the revisions did not offer the necessary solidity and feel that the manuscript cannot be accepted for publication in its present form.

Response: We greatly appreciate your valuable comments and constructive suggestion to our work. We have carefully revised the MS following your valuable suggestions and comments.

The main findings of our study include: 1) orphan salivary proteins might be originated from HGT; 2) the newly acquired genes undergo duplications and function differently through co-option evolution. Although the former evidence was not very strong, we provide solid evidence of diversified functions of newly acquired genes.

For the HGT results, we discuss the other possibility you mentioned in the revised version in Line 239-249: “However, compared with most reported HGT events that had a lot of homologues in donor relatives¹⁸, only a few bacteria species contain HESP-associated genes. Moreover, in current phylogenetic tree, insect HESPs did not cluster with bacterial genes in strongly supported bootstrap (Figure 4). Therefore, current evidence for HGT is not very strong. We cannot rule out the possibility that HESP-associated genes in bacteria and insects were independently evolved, and their sequence similarity might be resulted from convergent evolution. Also, there is a possibility that the HGT event was originated from insects to bacteria, as *Eutrichophora* species were intimately associated with symbiotic microorganism²⁹. Based on above results, our study only provided a possible origin of HESP. The exact phylogeny of HESP still needs further investigations. Perhaps a high-throughput sequencing of a wider range of organisms in the future might solve this issue”.

Reviewer #2 (Remarks to the Author):

This resubmitted manuscript is carefully revised and well presented. The revised discussion and methods are up to date, the analyses in the result is adequate. It seems much better than the last one. But I still have some minor comments.

Response: Thank you very much for your positive comments and constructive suggestion to our work. We have carefully revised the MS following your valuable suggestions and comments.

1) Authors focused on *Riptortus pedestris* and *Pyrrhocoris apterus* in most study, but only *Oncopeltus fasciatus* phylogenetical analysis was shown. The reason in the introduction (line75) appeared to be farfetched, because many important species are in those three superfamilies. I suggest authors explain the reasons in the Method section through referring to the phylogenetic relationship and useful genomic data of *Oncopeltus fasciatus*.

Response: Thank you for pointing out this. In the revised version, we addressed the reason for selecting these three species as “*Riptortus pedestris* (Coreoidea), *Pyrrhocoris apterus* (Pyrrhocoroidea), and *Oncopeltus fasciatus* (Lygaeoidea) are important species in these three superfamilies with available genome information” in Line 75, “Currently, the genomes of three insects in Coreoidea, Pyrrhocoroidea, and Lygaeoidea, respectively, are available, including *R. pedestris*, *P. apterus*, and *O. fasciatus*” in Methods and Introduction sections (Line 367-368). Up to now, except for these three species, no available genome was reported in Eutrichophora.

For the reason not involved in RNAi on *O. fasciatus*, we addressed that *O. fasciatus* is distributed in North America, and is not proceeded for functional analysis in this study in the Method section in Line 394-395 (we cannot rear *O. fasciatus* in our lab).

2) Even author updated and used the new name HESP in the whole manuscript, there are still some ESSP in figure 1, figure 2 and figure 6. Please change all ESSP to HESP in the figures.

Response: Sorry about these mistakes. In the revised version, we carefully check the manuscript, and these errors have been corrected.

3) Line 638 should be “Effects of dsPaHESP treatments on *Pyrrhocoris apterus*”.

Response: Thank you for pointing out this. We corrected it in the revised version.

4) Figure 3, the name of bacteria protein should not be HESP. Because authors defined HESP as horizontal-transferred, Eutrichophora-specific salivary protein.

Response: Thank you for your suggestion. In the revised version, the gene name of bacteria was changed, and the NCBI accession was provided following the “protein” in the figure. Also, the use of bacteria gene name was carefully checked in the manuscript. Thank you for all your helps in improving our manuscript.